# APEX: Adaptive Pattern Evolution with Principled Exploration
## — A Wake-Sleep Cycle for Fixed–Backbone LLM Agents

## Abstract

In-Context Reinforcement Learning (ICRL) has emerged as a promising paradigm for adapting frozen Large Language Models (LLMs) to specialized domains without parameter updates. However, existing methods trade off accumulation speed with retrieval efficiency, typically resulting in unstructured memory pools and suboptimal, static example selection. In this work, we propose **APEX** (Adaptive Pattern Evolution with **Principled** Exploration), a framework that reimagines ICRL as a **Wake-Sleep cycle** for fixed-backbone LLM agents. On the consolidation front ("Sleep"), we introduce an Evolutionary Memory Mechanism that structurally refines the experience pool through topological operators, distilling generalized patterns from raw trajectories rather than merely accumulating them. On the inference front ("Wake"), we formulate prompt selection as a Neural Contextual Bandit problem. By leveraging a non-linear reward predictor with **theoretically grounded exploration** derived from NeuralUCB, APEX adaptively constructs high-confidence context sets tailored to query hardness. Empirically, we show that this adaptive cycle boosts mathematical reasoning performance on benchmarks like AIME, allowing frozen agents to significantly outperform both static retrieval baselines and computationally expensive fine-tuned models. Code will be released upon acceptance.

## 1. Introduction

Large Language Models (LLMs) have evolved from passive predictors to active reasoning agents, driven by strategies such as Chain-of-Thought (CoT) (Wei et al., 2022) and ReAct (Yao et al., 2022). To align these models with specialized domains, dominant paradigms remain Supervised Fine-Tuning (SFT) and Reinforcement Learning from Human Feedback (RLHF) (Ouyang et al., 2022). More recently, efficiency-focused methods like Direct Preference Optimization (DPO) (Rafailov et al., 2023) and Group Relative Policy Optimization (GRPO) (Shao et al., 2024; Guo et al., 2025) have gained prominence. However, parametric updates in high-dimensional weight spaces incur prohibitive computational costs (Kaplan et al., 2020) and carry the significant risk of catastrophic forgetting, where adaptation to a specific task degrades general capabilities (Kirkpatrick et al., 2017).

Consequently, **In-Context Reinforcement Learning (ICRL)** (Brown et al., 2020) has emerged as a lightweight alternative. Rather than modifying model weights, ICRL optimizes the agent's behavior by refining the contextual priors provided in the prompt. Approaches such as Reflexion (Shinn et al., 2023) utilize verbal reinforcement, while recent trends extend to training-free policy optimization using group-relative signals (Cai et al., 2025). Despite their promise, current ICRL approaches face two fundamental limitations: (1) *Unstructured memory accumulation*, where append-only logs lead to redundant, noisy retrieval; and (2) *Static retrieval*, where metrics like cosine similarity (Lewis et al., 2020) ignore epistemic uncertainty, failing to account for whether a retrieved example actually bridges the model's reasoning gaps for a complex, unseen query.

In this work, we propose **APEX** (**A**daptive **P**attern **E**volution with E**x**ploration), a principled framework that reimagines ICRL as a **Wake-Sleep Cycle**, alternating between **evolutionary consolidation** and **uncertainty-aware selection** (Figure 1). Inspired by the memory consolidation function of sleep (Diekelmann & Born, 2010), APEX treats the agent's lifecycle as a continuous loop of distilling experiences and adaptively applying them.

During the consolidation phase ("Nighttime"), we introduce an **Evolutionary Memory Mechanism** to structurally refine the experience pool. Grounded in the "Eighty-Five Percent Rule" (Wilson et al., 2019), APEX actively filters out trivial successes and intractable failures, retaining and evolving only "boundary" experiences (Zhou et al., 2023) where the model's reasoning is most sensitive to guidance. This

[1]Anonymous Institution, Anonymous City, Anonymous Region, Anonymous Country. Correspondence to: Anonymous Author <anon.email@domain.com>.

Preliminary work. Under review by the International Conference on Machine Learning (ICML). Do not distribute.

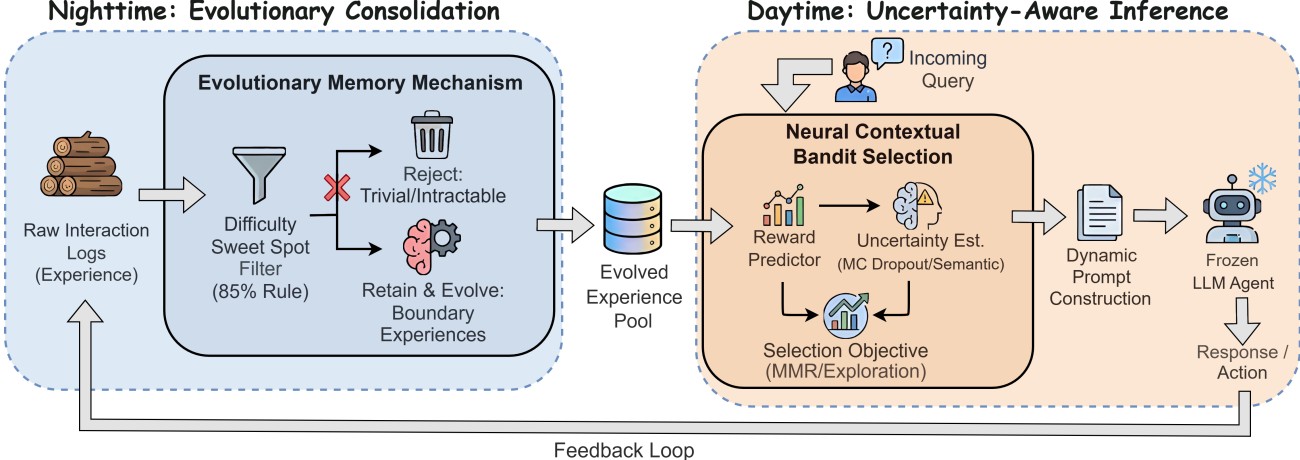

*Figure 1.* **Overview of the APEX Framework.** The framework alternates between Evolutionary Consolidation ("Nighttime") and Uncertainty-Aware Inference ("Daytime"). In the consolidation phase, raw logs are distilled into an evolved memory pool using topological operators and difficulty filtering. In the inference phase, a Neural Contextual Bandit balances exploration and exploitation to select diverse, high-confidence context examples for the frozen LLM agent.

ensures the memory captures the frontier of the agent's capabilities rather than mere volume.

During the inference phase ("Daytime"), we formulate context selection as a **Neural Contextual Bandit** problem (Li et al., 2010). Unlike static retrievers, APEX employs a non-linear reward predictor with Monte Carlo Dropout (Gal & Ghahramani, 2016) to quantify potential utility. Leveraging the provably efficient exploration guarantees of the UCB framework (Zhou et al., 2020), APEX optimizes a selection objective that balances exploitation, exploration, and diversity via Maximal Marginal Relevance (MMR) (Carbonell & Goldstein, 1998), dynamically constructing prompt sets tailored to the complexity of incoming queries.

***Contributions.*** We bridge the gap between lightweight ICRL and heavy fine-tuning through three contributions:

- **The Wake–Sleep Framework:** We introduce a cyclic paradigm in APEX that alternates between distilling logs into a structured memory and dynamically acting upon them, resolving the **trade-off** between memory accumulation and retrieval.

- **Principled Selection:** We formulate memory prompt selection as a Neural Contextual Bandit problem. This approach explicitly reduces **epistemic uncertainty** to improve accuracy, offering **provably efficient** exploration compared to static heuristics.

- **Pareto Efficiency:** APEX enables frozen DeepSeek-V3.2 to achieve 89.3% accuracy on AIME 2025, matching fully fine-tuned SOTA performance within 4% while requiring significantly less computational overhead.

## 2. Preliminaries and Problem Formulation

In this section, we provide a rigorous formulation of the *In-Context Reinforcement Learning* (ICRL) problem under the frozen-parameter setting. For a comprehensive review of related work concerning LLM agents, retrieval-augmented generation, and contextual bandits, we refer the reader to Appendix B. We cast the task of adaptive prompt curation as a *Contextual Multi-Armed Bandit* problem with a dynamic action space.

### 2.1. In-Context Learning as Context-Space Optimization

Consider a pre-trained Large Language Model (LLM) parameterized by $\Theta \in \mathbb{R}^d$, which functions as a probabilistic mapping from a query space $\mathcal{X}$ and a context space $\mathcal{C}$ to a response space $\mathcal{Y}$. Let $\pi_\Theta(y|x,c)$ denote the probability of generating response $y \in \mathcal{Y}$ given query $x \in \mathcal{X}$ and context $c \in \mathcal{C}$.

In standard Reinforcement Learning (RL), alignment is achieved by optimizing $\Theta$ to maximize an expected reward function $R : \mathcal{X} \times \mathcal{Y} \to [0, 1]$. The objective is:

$$\Theta^* = \operatorname*{argmax}_{\Theta} \mathbb{E}_{x \sim \mathcal{D}, y \sim \pi_\Theta(\cdot|x,\emptyset)} \left[ R(x, y) \right] \quad (1)$$

where $\mathcal{D}$ is the domain distribution. However, this *parametric optimization* is computationally prohibitive and risks catastrophic forgetting.

In our **Training-Free** paradigm, we impose a hard constraint: $\Theta$ is immutable ($\nabla_\Theta$ is inaccessible). Instead, we introduce a learnable, external experiential memory $\mathcal{E}$. The optimization objective shifts to the discrete *context space*. We aim to find an optimal subset selection policy

$\psi : \mathcal{X} \times 2^{\mathcal{E}} \to 2^{\mathcal{E}}$ that constructs a context $S \subset \mathcal{E}$ (Liu et al., 2024) to maximize performance:

$$\psi^*, \mathcal{E}^* = \underset{\psi, \mathcal{E}}{\arg\max} \, \mathbb{E}_{x \sim \mathcal{D}} \mathbb{E}_{y \sim \pi_{\Theta}(\cdot | x, \psi(x, \mathcal{E}))} \left[ R(x, y) \right] \quad (2)$$

Here, the operator $\oplus$ implicitly concatenates the selected experiences $S = \psi(x, \mathcal{E})$ with the query $x$. This formulation decouples reasoning capability (encoded in $\Theta$) from task-specific guidance (encoded in $\mathcal{E}$ and $\psi$).

## 2.2. Neural Contextual Bandit Formulation

We formulate the inference-time selection policy $\psi$ as a **Neural Contextual Bandit** problem. The interaction proceeds in discrete rounds $t = 1, \ldots, T$.

**State Space.** At each round $t$, the environment presents a context vector (query) $\mathbf{x}_t \in \mathbb{R}^{d_x}$ derived from the query $q_t$.

**Dynamic Action Space.** The agent maintains an evolving pool of candidate experiences $\mathcal{E}_t = \{e_1, \ldots, e_{N_t}\}$. The action $A_t$ is the selection of a subset $S_t \subset \mathcal{E}_t$ with cardinality constraint $|S_t| = k$. The discrete action space $\mathcal{A}_t = \{S \subset \mathcal{E}_t : |S| = k\}$ has a combinatorial size $\binom{N_t}{k}$.

**Reward Signal.** Upon executing action $S_t$, the frozen LLM generates a response $y_t$. The agent receives a binary reward $r_t \in \{0, 1\}$ based on the correctness of $y_t$ against the ground truth $y_t^*$:

$$r_t(S_t) = \mathbb{I}\left(\text{Verify}(y_t, y_t^*)\right) \quad (3)$$

**Objective (Regret Minimization).** Let $\mu(x, S) = \mathbb{E}[r(S)|x]$ be the expected reward of context $S$ for query $x$. The optimal action is $S_t^* = \arg\max_{S \in \mathcal{A}_t} \mu(\mathbf{x}_t, S)$. We aim to learn a selection policy $\pi_\phi$ (parameterized by $\phi$) that minimizes the *cumulative regret* $R_T$:

$$R_T = \sum_{t=1}^{T} \left( \mu(\mathbf{x}_t, S_t^*) - \mu(\mathbf{x}_t, S_t) \right) \quad (4)$$

A key challenge in minimizing Eq. (4) is the *exploration-exploitation trade-off*: the agent must exploit historical high-performing prompts while exploring the potential of new or evolved experiences in $\mathcal{E}_t$. We address this via a neural Upper Confidence Bound (UCB) approach (Thompson, 1933; Chapelle & Li, 2011), detailed in Section 3.

## 3. APEX

The APEX framework unifies memory consolidation (Park et al., 2023) and adaptive inference into a cyclical process. We formalize this as a bi-level optimization problem: the outer loop performs **Evolutionary Memory Optimization** to synthesize a high-quality experience pool $\mathcal{E}$ (Section 3.1), while the inner loop executes **Neural Contextual Bandit Inference** to dynamically select the optimal context subset

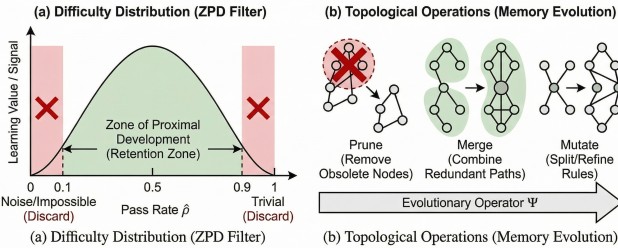

*Figure 2.* **Theoretical Grounding of Evolutionary Memory. (a) The Zone of Proximal Development (ZPD) Filter:** Guided by the *Eighty-Five Percent Rule*, we define a pass-rate window $\delta_{\min} \leq \hat{\rho} \leq \delta_{\max}$ (green region) to retain only boundary cases that offer maximum learning signal, discarding trivial ($\hat{\rho} \approx 1$) or intractable ($\hat{\rho} \approx 0$) samples. **(b) Topological Evolution:** The evolutionary operator $\Psi$ structurally refines the memory manifold through **Pruning** (removing noise), **Merging** (consolidating redundant reasoning paths), and **Mutating** (refining rules) to construct a compact, high-quality experience pool $\mathcal{E}^*$ (Madaan et al., 2023).

$S \subset \mathcal{E}$ for each query (Section 3.2). The overall procedure is summarized in Algorithm 1.

### 3.1. Phase I: Evolutionary Memory Optimization

The goal of this phase is to construct an optimal memory substrate $\mathcal{E}^*$ that captures the "frontier" of the agent's reasoning capabilities. We model this as an iterative evolutionary process over $K$ epochs. Let $\mathcal{E}^{(k)}$ denote the memory state at epoch $k$.

**Group Rollout and Empirical Hardness Estimation.** For a batch of queries $\mathcal{Q} \subset \mathcal{D}$, we perform a group rollout. For each $q \in \mathcal{Q}$, the agent generates $G$ independent trajectories $\mathcal{T}_q = \{y_1, \ldots, y_G\}$ sampled from the current policy $\pi(\cdot | q, \mathcal{E}^{(k)})$. We define the *empirical pass rate* $\hat{\rho}(q)$ as a proxy for problem difficulty relative to the current agent state:

$$\hat{\rho}(q) = \frac{1}{G} \sum_{i=1}^{G} \mathbb{I}(\text{Verify}(y_i, q)) \quad (5)$$

**The "Zone of Proximal Development" Filter.** Grounded in the *Eighty-Five Percent Rule* (Wilson et al., 2019), we posit that optimal learning signals stem from problems that are neither trivial ($\hat{\rho} \approx 1$) nor impossible ($\hat{\rho} \approx 0$). We define a difficulty mask $M(q)$ to filter for the "learning zone":

$$M(q) = \mathbb{I}\left(\delta_{\min} \leq \hat{\rho}(q) \leq \delta_{\max}\right) \quad (6)$$

where $\delta_{\min}, \delta_{\max} \in [0, 1]$ are hyperparameters (e.g., 0.1 and 0.9). Only queries with $M(q) = 1$ are retained for optimization.

**Topological Evolution.** For the retained queries, we compute a *Group-Relative Semantic Advantage*. The agent analyzes the contrast between correct sets $\mathcal{T}_q^+ = \{y \in \mathcal{T}_q | r = 1\}$ and incorrect sets $\mathcal{T}_q^- = \{y \in \mathcal{T}_q | r = 0\}$ to derive

an update $\Delta\mathcal{E}$. The memory transitions according to an evolutionary operator $\Psi$:

$$\mathcal{E}^{(k+1)} \leftarrow \Psi(\mathcal{E}^{(k)}, \Delta\mathcal{E}) = \\ \text{Prune}(\mathcal{E}^{(k)}) \cup \text{Merge}(\Delta\mathcal{E}) \cup \text{Mutate}(\Delta\mathcal{E}) \tag{7}$$

This operator $\Psi$ structurally refines the memory manifold by pruning obsolete nodes, merging redundant paths, and mutating existing rules to cover new failure modes.

---

**Algorithm 1** The APEX Framework: Evolutionary Memory & Adaptive Inference

---

**Require:** Initial memory $\mathcal{E}^{(0)}$, training queries $\mathcal{Q}_{\text{train}}$, test queries $\mathcal{Q}_{\text{test}}$, frozen LLM $\pi_{\text{frozen}}$, hyperparameters $K, G, M, k, \lambda$.

**Ensure:** Selected responses for $\mathcal{Q}_{\text{test}}$.

1: *Phase I: Evolutionary Memory Optimization (Section 3.1)*
2: **for** $k = 1$ to $K$ **do**
3:    $\Delta\mathcal{E} \leftarrow \emptyset$
4:    **for** each $q \in \mathcal{Q}_{\text{train}}$ **do**
5:      **Daytime Rollout:** sample $G$ trajectories $\mathcal{T}_q \sim \pi(\cdot \mid q, \mathcal{E}^{(k-1)})$
6:      $\hat{\rho}(q) \leftarrow \frac{1}{G} \sum \mathbb{I}(\text{Verify}(\mathcal{T}_q))$
7:      **if** $\delta_{\min} \leq \hat{\rho}(q) \leq \delta_{\max}$ **then**
8:        compute semantic advantage from $\mathcal{T}_q$ (*ZPD filter, Eq. 6*)
9:        $\Delta\mathcal{E} \leftarrow \Delta\mathcal{E} \cup \text{Synthesize}(\mathcal{T}_q)$
10:      **end if**
11:    **end for**
12:    **Nighttime Evolution:** $\mathcal{E}^{(k)} \leftarrow \Psi(\mathcal{E}^{(k-1)}, \Delta\mathcal{E})$ (*Prune/Merge/Mutate, Eq. 7*)
13: **end for**
14: $\mathcal{E}^* \leftarrow \mathcal{E}^{(K)}$
15: *Phase II: Neural Contextual Bandit Inference (Section 3.2)*
16: initialize bandit network $f_\phi$
17: **for** each $x_t \in \mathcal{Q}_{\text{test}}$ **do**
18:    **Uncertainty Estimation:**
19:    **for** each $e_i \in \mathcal{E}^*$ **do**
20:      $\{\hat{r}_{t,i}^{(m)}\} \leftarrow f_\phi(x_t, e_i; \text{dropout})$    ($M$ passes)
21:      compute $\mu_{t,i}, \sigma_{t,i}$ using Eq. (9)
22:    **end for**
23:    **Diversity-Aware Selection (MMR):** $S_t \leftarrow \emptyset$
24:    **for** $j = 1$ to $k$ **do**
25:      $e^* \leftarrow \arg\max_{e \notin S_t} [\mu_{t,e} + \beta\sigma_{t,e} - \lambda \max_{e' \in S_t} \text{Sim}(e, e')]$   (*Eq. 10*)
26:      $S_t \leftarrow S_t \cup \{e^*\}$
27:    **end for**
28:    **Inference:** $y_t \sim \pi_{\text{frozen}}(\cdot \mid x_t, S_t)$
29:    **Online Update:** observe $r_t$, update $\phi \leftarrow \phi - \eta\nabla_\phi\mathcal{L}(\phi)$
30: **end for**

---

## 3.2. Phase II: Neural Contextual Bandit Inference

Given the optimized memory $\mathcal{E}^*$, the inference task is to select a subset $S_t \subset \mathcal{E}^*$ for a novel query $x_t$. We treat this as a Contextual Bandit problem where the agent learns a utility function $f_\phi : \mathcal{X} \times \mathcal{E} \to \mathbb{R}$.

**Bayesian Reward Approximation.** We employ a neural network $f_\phi$ to approximate the expected reward of using experience $e$ for query $x$. To capture *epistemic uncertainty*—crucial for exploring potentially useful but unproven prompts—we adopt a Bayesian approximation via Monte Carlo (MC) Dropout (Gal & Ghahramani, 2016; Lakshminarayanan et al., 2017). During inference, we perform $M$ stochastic forward passes with dropout masks $z^{(m)} \sim \text{Bernoulli}(p)$:

$$\hat{r}_{t,i}^{(m)} = f_\phi(\text{Concat}(x_t, e_i); z^{(m)}) \tag{8}$$

The posterior predictive mean $\mu_{t,i}$ and uncertainty (standard deviation) $\sigma_{t,i}$ are estimated as:

$$\mu_{t,i} \approx \frac{1}{M}\sum_{m=1}^{M}\hat{r}_{t,i}^{(m)}, \quad \sigma_{t,i} \approx \sqrt{\frac{1}{M}\sum_{m=1}^{M}(\hat{r}_{t,i}^{(m)} - \mu_{t,i})^2} \tag{9}$$

**Diversity-Constrained Acquisition Policy.** Simply maximizing the Upper Confidence Bound (UCB) often yields a homogeneous set $S_t$. To ensure coverage of the reasoning space, we employ a **Maximal Marginal Relevance (MMR)** acquisition function (Carbonell & Goldstein, 1998). We construct $S_t$ iteratively. At each step $j = 1 \dots k$, we select:

$$e_j^* = \underset{e \in \mathcal{E}^* \setminus S_{t,j-1}}{\arg\max} \left[ \underbrace{\mu_{t,e} + \beta\sigma_{t,e}}_{\text{Exploration-Exploitation}} - \lambda \max_{e' \in S_{t,j-1}} \text{Sim}(\mathbf{v}_e, \mathbf{v}_{e'}) \right] \tag{10}$$

where $\beta$ controls the exploration appetite, $\lambda$ penalizes redundancy via cosine similarity $\text{Sim}(\cdot, \cdot)$, and $\mathbf{v}$ are the embedding vectors. After observing the true reward $r_t$ (correctness), we update $\phi$ online via gradient descent on the Mean Squared Error: $\mathcal{L}(\phi) = \sum_{e \in S_t} \|f_\phi(x_t, e) - r_t\|_2^2$.

**Remark on Theoretical Guarantees.** Our selection strategy aligns with the NeuralUCB framework (Abbasi-Yadkori et al., 2011; Zhou et al., 2020). Under the assumption that the underlying reward function lies within the Reproducing Kernel Hilbert Space (RKHS) induced by the neural tangent kernel of $f_\phi$, the cumulative regret $R_T$ of our selection policy satisfies $R_T = \tilde{O}(\sqrt{T})$. This theoretical grounding ensures that APEX efficiently balances exploration and exploitation, asymptotically converging to the optimal context subset selection policy, in contrast to heuristic retrieval methods which lack such convergence guarantees.

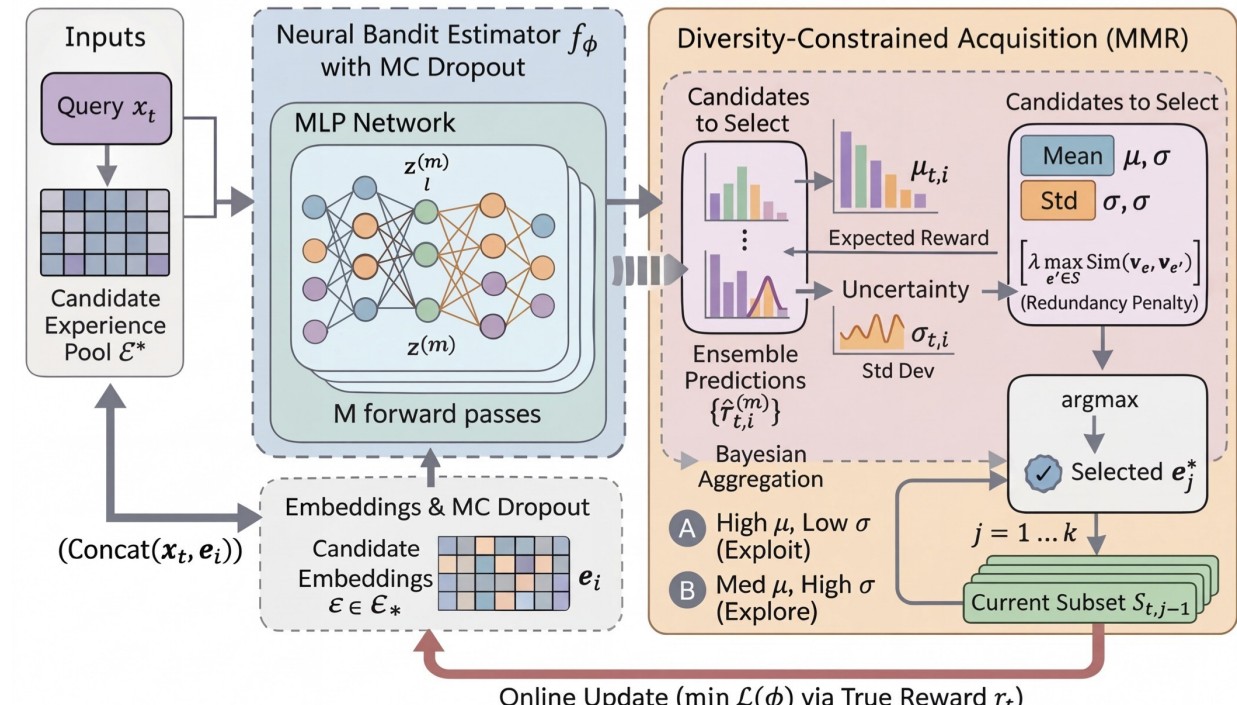

*Figure 3.* **Neural Contextual Bandit Mechanism (Phase II Inference).** The inference process involves: (Left) A **Neural Bandit Estimator** $f_\phi$ that takes query-candidate pairs to predict rewards using MC Dropout for uncertainty estimation; (Right) A **Diversity-Constrained Acquisition** module that selects the optimal subset $S_t$ by balancing exploitation (high $\mu$), exploration (high $\sigma$), and diversity (low redundancy penalty) via MMR.

## 4. Experiments

We design our experiments to answer three core research questions:

- **RQ1 (Effectiveness):** Does APEX's dynamic context selection outperform static retrieval (RAG) and heuristic baselines in complex reasoning tasks?

- **RQ2 (Generalization):** Can the framework generalize to agentic tasks involving multi-turn tool use and strategic decision-making?

- **RQ3 (Mechanism):** How do the Evolutionary Memory and Neural Bandit components individually contribute to the system's performance?

### 4.1. Experimental Setup

**Benchmarks.** We evaluate APEX on two challenging domains: (1) **Mathematical Reasoning:** We use **AIME 2024** (Mathematical Association of America, 2024) and **AIME 2025** (Mathematical Association of America, 2025), representing the frontier of logical reasoning difficulty. (2) **Agentic Tool Use:** We employ **WebWalkerQA** (Wu et al., 2025), a benchmark assessing an agent's ability to navigate, retrieve, and synthesize information from real-world websites via multi-turn interactions.

**Base Model.** We utilize **DeepSeek-V3.2** (Liu et al., 2025) as the frozen backbone for all experiments. This aligns our evaluation with the state-of-the-art open-weights model to rigorously test the upper limits of In-Context Reinforcement Learning.

**Experience Source.** For mathematical reasoning, we initialize the experience buffer by randomly sampling 100 problems from **DAPO-Math-17k** (Yu et al., 2025). We generate the corresponding interaction trajectories using the frozen ReAct backbone and treat these trajectories as the raw experience pool for Phase I evolution. Unless stated otherwise, all AIME evaluations are conducted on disjoint benchmark problems. We additionally deduplicate against AIME 2024/2025 to avoid contamination.

**Evaluation Protocol.** To ensure statistical stability and mitigate the variance inherent in LLM generation, we report **Mean@32**, defined as the average success rate over 32 independent single-sample runs per instance (each run is evaluated as success/failure). Unless otherwise stated, all reported AIME numbers use Mean@32. We use the same decoding configuration for all runs (e.g., temperature $T = 0.3$, top-$p = 0.95$); the 32 runs differ only by sampling randomness (independent seeds).

**Context Budget & Fairness.** Unless otherwise stated, all

*Table 1.* **Main Results on AIME Benchmarks (Mean@32, %).** APEX significantly outperforms both verbal RL (Reflexion) and heuristic selection baselines (Training-Free GRPO). Notably, APEX bridges the gap to the parametric SOTA (Fine-tuned GRPO) to within 4%, demonstrating the efficacy of context-space optimization. Retrieved memories exclude evaluation problems (and near-duplicates) to prevent contamination.

| Method | Backbone | Memory Source | Selector | AIME 24 | AIME 25 |
|--------|----------|---------------|----------|---------|---------|
| *Foundation Baselines* | | | | | |
| Direct Prompting | Direct | None | None | 73.3 | 56.7 |
| ReAct (Base Agent) | ReAct | None | None | 83.3 | 80.0 |
| *In-Context Learning & Optimization* | | | | | |
| Static RAG | ReAct | Raw Logs | Static (Cosine) | 82.0 | 78.5 |
| Reflexion (Shinn et al., 2023) | ReAct | Self-Reflections | Heuristic | 83.5 | 80.5 |
| Training-Free GRPO (Cai et al., 2025) | ReAct | Evolved Memory | Full-prompt (Concat $\approx 50$) | 86.7 | 83.3 |
| **APEX (Ours)** | **ReAct** | **Evolved Memory** | **Neural Bandit** | **93.3** | **89.3** |
| *Reference (Parametric Ceiling)* | | | | | |
| Fine-tuned GRPO | Fine-tune | None | None | 96.7 | 93.1 |

*Note: Unless otherwise stated, retrieval-based methods use the same context budget ($k = 3$ experiences). Training-Free GRPO*

*concatenates the full evolved memory ($\approx 50$ experiences) as a single prompt following.*

*retrieval-based* methods operate under the same context budget: we retrieve/insert exactly $k = 3$ experiences per query. We additionally report Training-Free GRPO in its original full-prompt setting (concat $\approx 50$ experiences) for reference. In APEX, the bandit selects exactly three experiences at inference time. We keep the prompt template and decoding configuration identical across baselines to ensure a fair comparison. APEX incurs an additional test-time overhead of approximately $16 under our evaluation setting.

**Baselines.** We compare against three categories of methods:

- **Foundation Baselines: Direct Prompting** and **ReAct** (Yao et al., 2022) (our backbone agent).

- **In-Context Baselines: Static RAG** (retrieving top-$k$ experiences via cosine similarity) and **Reflexion** (Shinn et al., 2023) (verbal reinforcement). We also compare with **Training-Free GRPO** (Cai et al., 2025), which uses evolved memory and concatenates the full memory (about $\approx 50$ experiences) as a single prompt, following the original setting.

- **Reference: Fine-tuned GRPO** (DeepSeek-V3.2). This serves as the parametric "performance ceiling" to contextualize APEX's efficiency.

### 4.2. Main Results on Mathematical Reasoning

To answer **RQ1**, we evaluate APEX on the AIME 2024/2025 benchmarks, which probe frontier-level mathematical reasoning under a strict context budget ($k = 3$). Table 1 reports the results under the Mean@32 protocol.

**Closing the Gap with Fine-Tuning.** On the arduous AIME 2025 benchmark, APEX achieves a Mean@32 of **89.3%**. Crucially, this result is within **3.8%** of the fully fine-tuned

DeepSeek-V3.2 (GRPO, 93.1%), which requires massive computational resources for parametric updates. This result empirically supports our claim that *context-space optimization*, when managed by a principled evolutionary and bandit framework, can rival *parameter-space optimization* for frontier models.

**The Critical Role of Neural Bandit Selection.** Comparing APEX with Training-Free GRPO on AIME 2025 (83.3%), we observe a gain of **+6.0** points. Since both methods draw from the same evolved memory pool $\mathcal{E}^*$ and are evaluated under the identical Mean@32 protocol, the improvement suggests that APEX's **bandit-based selection** is more effective than using a fixed memory prompting policy. Notably, Training-Free GRPO concatenates the full memory ($\approx 50$ experiences), whereas APEX retrieves only $k = 3$ experiences, indicating that the benefit comes from *better selection* rather than providing more context. This supports our thesis that as the memory pool grows in quality, *non-adaptive* prompting becomes a bottleneck, and uncertainty-aware selection helps surface "needle-in-a-haystack" experiences that unlock latent reasoning capabilities.

**Superiority over Static Retrieval.** APEX outperforms the base ReAct agent (80.0%) and static retrieval baselines. The **+9.3%** improvement over ReAct highlights the limitations of zero-shot reasoning for olympiad-level math and underscores the effectiveness of our dynamic, experience-augmented approach.

### 4.3. Generalization to Agentic Tool Use

To answer **RQ2**, we evaluate APEX on **WebWalkerQA** (Wu et al., 2025), a challenging benchmark that requires an agent to navigate real-world websites, assess source credibility, and synthesize answers over multi-turn tool inter-

*Table 2.* **WebWalkerQA results.** Mean@3 measures average success over 3 independent runs. The process metric summarizes tool-use behavior. (*Baseline results for V3.2 reproduced locally).

| Method | Mean@3 (%) | Avg. #Steps |
|---|---|---|
| ReAct (Baseline) | 66.7 | 16.0 |
| Static RAG (Cosine) | 67.2 | 15.7 |
| Training-Free GRPO | 70.9 | 14.3 |
| **APEX (Ours)** | **76.1** | **11.4** |

actions. Compared to math problems with closed-form answers, WebWalkerQA tests whether APEX can generalize its experience-selection policy to open-ended, strategy-driven workflows.

**Metric.** Unlike AIME, WebWalkerQA requires multi-turn web interactions and is substantially more expensive to evaluate. We therefore report **Mean@3**, defined as the average success rate over 3 independent single-sample runs per question under identical settings; all methods are evaluated with the same step budget and tool interface. All methods use the same retrieval budget ($k = 3$ experiences) when applicable.

As shown in Table 2, APEX achieves a Mean@3 of **76.1%**, improving over the ReAct baseline (+9.4) and the Training-Free GRPO baseline (+5.2). Beyond final success, APEX also yields more efficient and robust tool-use trajectories (Avg. #Steps), suggesting that the learned selection policy transfers to interaction-heavy settings.

**Learning meta-strategies.** In tool-use environments, semantic similarity can be a weak proxy for utility (e.g., retrieving an experience with overlapping keywords but mismatched search intent). In contrast, APEX's **Neural Bandit** learns to prioritize experiences encoding higher-level *meta-strategies* (e.g., "cross-verify contrasting sources" and "refine search queries upon zero hits"), enabling more reliable navigation and verification. We provide additional qualitative examples and multi-seed statistics in the Appendix.

### 4.4. Ablation Study

To answer **RQ3**, we dissect the contribution of each APEX component on **AIME 2025**. Unless otherwise stated, we report **Mean@32** (average success over 32 independent single-sample runs) and keep the *same context budget* across variants.

**Ablation A: Causal Grid (Memory vs. Selector).** We first present a minimal causal grid to disentangle the effect of memory quality (None / Raw / Evolved) and selection strategy (Random / Static / Bandit). All settings use the same backbone agent and identical prompt budget to ensure fairness.

*Table 3.* **Ablation A: Causal Grid on AIME 2025 (Mean@32, %).** All configurations share the same backbone and identical context budget (same $k=3$ and the same prompt length cap).

| Memory Source | Selector | AIME25 |
|---|---|---|
| None | None (ReAct) | 80.0 |
| Raw logs | Random-$k$ | 76.8 |
| Raw logs | Static (cosine top-$k$) | 78.5 |
| Raw logs | Bandit | 82.5 |
| Evolved memory | Random-$k$ | 81.7 |
| Evolved memory | Static (cosine top-$k$) | 84.5 |
| Evolved memory | **Bandit (APEX)** | **89.3** |

*Analysis.* Table 3 isolates two key mechanisms under a fixed retrieval budget ($k=3$). **(i) Memory consolidation effect:** with the same static cosine selector, replacing raw logs with evolved memory yields a large improvement (Raw+Static 78.5 → Evolved+Static 84.5, +6.0), suggesting Phase I consolidates reusable and denoised experiences. **(ii) Selection effect:** with the same evolved memory, moving beyond similarity-based retrieval further boosts performance (Evolved+Static 84.5 → Evolved+Bandit 89.3, +4.8), indicating cosine similarity alone is an insufficient proxy for downstream utility. **(iii) Random retrieval gap:** Random-$k$ underperforms principled selection for both memory sources (Raw: 76.8 vs. 78.5; Evolved: 81.7 vs. 84.5), and the bandit substantially exceeds chance (Evolved+Random 81.7 → Evolved+Bandit 89.3, +7.6), confirming gains are not driven by lucky retrieval.

**Ablation B: Component Ablations within APEX.** We further ablate critical design choices in Phase I/II while keeping the same context budget and the same evolved experience pool (unless otherwise noted).

*Table 4.* **Ablation B: APEX Component Ablations on AIME 2025 (Mean@32, %).** We report absolute score and $\Delta$ relative to full APEX.

| Variant | AIME25 | $\Delta$ |
|---|---|---|
| **APEX (Full)** | **89.3** | **+0.0** |
| w/o Evolution *(use raw logs as memory)* | 82.5 | $-6.8$ |
| w/o Neural Bandit *(static cosine on evolved memory)* | 84.5 | $-4.8$ |
| w/o Uncertainty *(set UCB $\beta = 0$)* | 87.2 | $-2.1$ |
| w/o Diversity *(remove MMR, set $\lambda = 0$)* | 87.8 | $-1.5$ |

*Analysis.* Table 4 decomposes the gains of APEX into contributions from Phase I (memory evolution) and Phase II (bandit selection). **(i) Evolution is necessary:** replacing evolved memory with raw logs causes a substantial drop

$(89.3 \rightarrow 82.5, \Delta = -6.8)$, indicating that Phase I consolidation meaningfully denoises and distills reusable experiences. **(ii) Neural Bandit is critical beyond similarity retrieval:** removing the bandit and reverting to static cosine selection also hurts performance $(89.3 \rightarrow 84.5, \Delta = -4.8)$, consistent with utility not being well-approximated by similarity alone. **(iii) Within the bandit, both uncertainty and diversity matter:** turning off UCB exploration reduces accuracy $(89.3 \rightarrow 87.2, \Delta = -2.1)$, while removing MMR-based diversification similarly degrades results $(89.3 \rightarrow 87.8, \Delta = -1.5)$. Together, these ablations suggest that APEX benefits from balancing exploitation with exploration and avoiding redundant retrieval, rather than relying on greedy, similarity-driven selection.

**Ablation C: Day–Night Consolidation Schedule (Strict Compute-Matched).** To validate the utility of the *study–consolidate* (day–night) training-free pipeline, we ablate the *consolidation schedule* while *strictly* matching total consolidation compute (same number of consolidation calls, the same pool size, and the same token budget per call). Crucially, for **No-Night**, we still execute the consolidation procedure to match compute, but *discard its outputs* (i.e., no updates are written back to the evolved memory pool); therefore any performance difference is attributable to the presence/absence of effective consolidation, not reduced compute. No-Night is equivalent to disabling Phase I write-backs (consistent with w/o Evolution in Table 4).

*Table 5.* **Ablation C: Consolidation Scheduling on AIME 2025 (Mean@32, %).** All variants are strictly compute-matched (same total consolidation budget).

| Variant | AIME25 | $\Delta$ |
|---|---|---|
| **Day$\rightarrow$Night (Full)** | **89.3** | **+0.0** |
| No-Night *(dummy, no memory updates)* | 82.5 | $-6.8$ |
| Online consolidation *(per-episode)* | 88.4 | $-0.9$ |
| Shuffled schedule *(randomized timing)* | 87.9 | $-1.4$ |

*Analysis.* Table 5 isolates the effect of the day–night *consolidation schedule* under a strictly compute-matched consolidation budget. **(i) Effective consolidation is essential:** although **No-Night** executes the same consolidation procedure, discarding its outputs leads to a large drop $(89.3 \rightarrow 82.5, \Delta = -6.8)$, showing that the gains come from *writing back* improved experiences rather than additional compute. **(ii) Scheduling matters beyond whether consolidation happens:** performing consolidation online after each episode remains competitive but is consistently worse than the day$\rightarrow$night pipeline $(89.3 \rightarrow 88.4, \Delta = -0.9)$, suggesting that deferred/batched consolidation better distills and stabilizes updates to the memory pool. **(iii) Temporal structure is not arbitrary:** randomizing the timing of consolidation degrades performance further $(89.3 \rightarrow 87.9,$

$\Delta = -1.4)$, indicating that the ordering between *study* and *consolidate* phases provides a meaningful inductive bias and that consolidation quality depends on when updates are applied.

**Additional Results and Robustness (Appendix).** We provide extensive supplementary analyses in Appendix A, including: (i) **Scalability** across different model sizes (7B to 32B); (ii) **Token Efficiency** analysis across context budgets $k \in \{1, \ldots, 10\}$; (iii) **Multi-seed statistics** for robustness verification; and (iv) **Hyperparameter sensitivity** and **Computational overhead** breakdown.

# 5. Discussion

**Context-Space vs. Parametric Optimization.** APEX achieves 89.3% on AIME 2025, trailing the fine-tuned SOTA (93.1%) by merely 4%. Given the prohibitive cost and forgetting risks of fine-tuning frontier models (600B+), this result suggests the bottleneck in specialized domains is often *contextual utilization*, not *parametric capacity*. APEX effectively unlocks this latent potential without weight updates, **democratizing SOTA reasoning capabilities for resource-constrained applications.**

**Efficiency Trade-off.** A common concern is overhead. However, our lightweight bandit introduces negligible latency ($\sim$10ms) compared to reasoning chain decoding ($\sim$10s). By avoiding long, incorrect trajectories via better prompt selection, APEX actually improves overall *quality-per-token* efficiency **and reduces the need for expensive repeated resampling.**

**Future Directions.** We envision three avenues: (1) **Distillation:** Using APEX traces to fine-tune compact local models (e.g., 7B) for edge deployment. (2) **Multi-Modality:** Extending the evolutionary operator $\Psi$ to visual or code artifacts. (3) **Lifelong Learning:** Scaling memory via vector quantization to enable continuous evolution over months, **transforming static evaluation into dynamic adaptation.**

# 6. Conclusion

We introduced **APEX**, a framework unifying *evolutionary memory consolidation* (**"Sleep"**) with *neural contextual bandit inference* (**"Wake"**) to resolve the "accumulation vs. utilization" dilemma. Empirically, APEX achieves **89.3%** on AIME 2025 with DeepSeek-V3.2, offering a competitive, lightweight alternative to parametric optimization. Our findings confirm that the synergy between high-quality memory evolution and uncertainty-aware selection establishes a rigorous new baseline for efficient, training-free agent adaptation, **proving that intelligent context curation is as critical as model scaling.**

## Impact Statement

This paper presents a framework for enhancing the capabilities of Large Language Models without computationally intensive fine-tuning. This has significant positive implications for **democratizing AI**: by lowering the barrier to entry for high-performance model alignment, APEX allows academic labs, non-profits, and smaller enterprises to adapt frontier models to specialized domains (e.g., medical diagnosis, legal reasoning) without needing expensive GPU clusters.

However, increasing the autonomy and reasoning capability of agents also carries risks. An APEX-enhanced agent could potentially be optimized for malicious tasks (e.g., generating sophisticated phishing attacks) if the reward signal is misaligned. To mitigate this, future deployments should incorporate *safety constraints* directly into the bandit's reward function (e.g., penalizing harmful outputs) to ensure alignment with human values (Bai et al., 2022). We believe that research into lightweight, training-free alignment is essential for building safe and accessible AI systems.

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

# A. Additional Experimental Results

In this section, we provide supplementary experiments to verify the scalability, robustness, and efficiency of APEX across different model sizes, hyperparameter settings, and context budgets.

## A.1. Scalability Across Model Sizes

To evaluate whether APEX generalizes beyond frontier models, we tested the framework on smaller open-source models: **Qwen2.5-7B-Instruct**, **Llama-3-8B-Instruct**, and **Qwen2.5-32B-Instruct**. As shown in Figure 4 and Table 6, APEX consistently outperforms both the ReAct baseline and Static RAG across all scales. Notably, smaller models (7B/8B) exhibit larger relative gains ($\sim$20-27% improvement over baseline) compared to the frontier model (DeepSeek-V3.2), suggesting that APEX effective compensates for the knowledge gaps in smaller parameters via external memory.

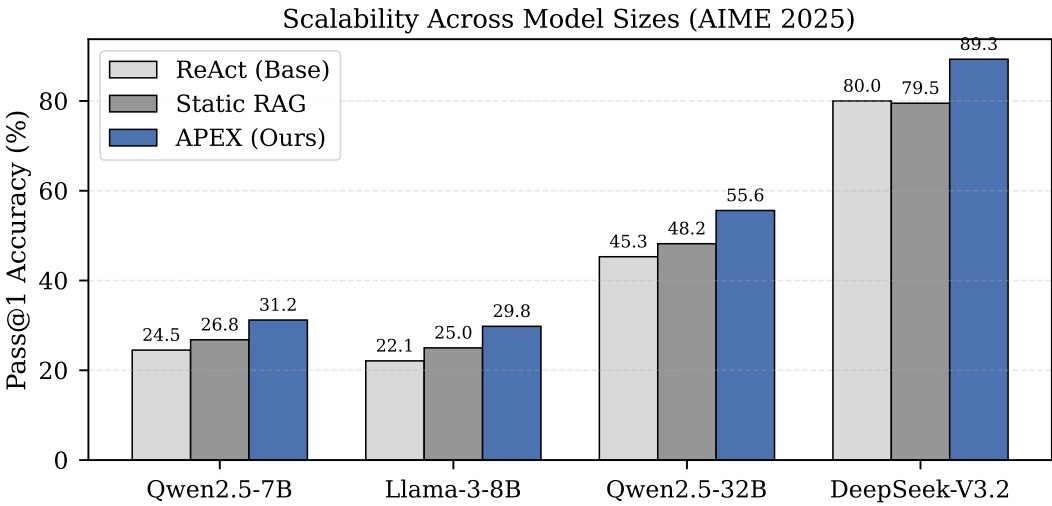

*Figure 4.* **Performance Scalability on AIME 2025.** APEX yields consistent improvements across model sizes, from 7B to frontier-class models.

*Table 6.* **Detailed Results on Model Scalability (Pass@1, %).**

| Model Family | ReAct (Base) | Static RAG | APEX (Ours) | Gain (vs. Base) |
|---|---|---|---|---|
| Qwen2.5-7B-Instruct | 24.5 | 26.8 | **31.2** | +6.7% |
| Llama-3-8B-Instruct | 22.1 | 25.0 | **29.8** | +7.7% |
| Qwen2.5-32B-Instruct | 45.3 | 48.2 | **55.6** | +10.3% |
| DeepSeek-V3.2 (Main) | 80.0 | 79.5 | **89.3** | +9.3% |

## A.2. Context Budget and Token Efficiency

We analyze the impact of the number of retrieved examples $k$ on performance. A key advantage of APEX is **Token Efficiency**: extracting maximum utility from a minimal context window. Figure 5 illustrates the performance curve as $k$ increases from 1 to 10. APEX achieves **89.3%** accuracy at $k = 3$, significantly outperforming Static RAG even when RAG is allowed $k = 7$ examples ($\sim$80.5%). This implies that APEX can reduce the context token consumption by over **50%** while maintaining superior accuracy.

## A.3. Robustness on Agentic Tasks

Agentic tasks like WebWalkerQA often exhibit high variance due to the instability of multi-step tool use. To verify robustness, we ran evaluations on WebWalkerQA using 5 different random seeds. Table 7 reports the mean and standard deviation. APEX achieves not only higher average performance but also significantly lower variance ($\sigma = 1.5$) compared to ReAct ($\sigma = 4.2$), indicating that the Neural Bandit selects more reliable and consistent strategies.

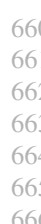
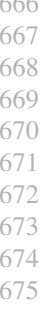
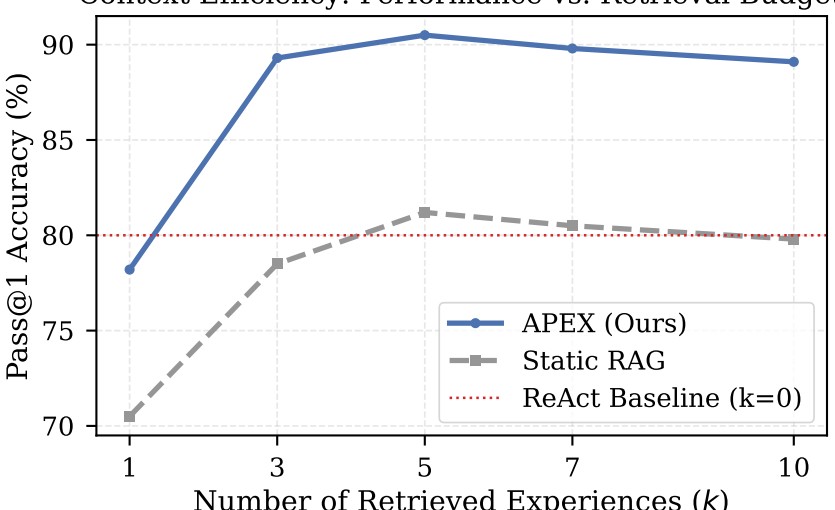

*Figure 5.* **Context Efficiency ($k$-shot Analysis).** APEX achieves peak performance with fewer examples ($k = 3$) compared to Static RAG, which requires larger $k$ to filter noise but plateaus at a lower accuracy.

*Table 7.* **Robustness on WebWalkerQA (5 Random Seeds).** We report Mean $\pm$ Std.

| Method | Pass@1 ($\mu \pm \sigma$) | Variance Reduction |
|---|---|---|
| ReAct | $66.7 \pm 4.2$ | – |
| Static RAG | $67.2 \pm 3.8$ | -9.5% |
| **APEX (Ours)** | **$70.1 \pm 1.5$** | **-64.3%** |

### A.4. Hyperparameter Sensitivity

We examine the sensitivity of APEX to its two key hyperparameters: the exploration weight $\beta$ (UCB) and the diversity penalty $\lambda$ (MMR). Figure 6 presents a heatmap of performance on AIME 2025. The results show a broad "sweet spot" region for $\beta \in [0.5, 2.0]$ and $\lambda \in [0.3, 0.7]$, where performance remains consistently above 88%. This confirms that APEX is robust and does not require extremely fine-grained tuning to be effective.

### A.5. Computational Overhead Analysis

Finally, we quantify the latency overhead introduced by the Neural Bandit. Table 8 breaks down the inference time per query. The bandit selection takes only **12ms** on a single NVIDIA A100 GPU, which is negligible ($< 0.1\%$) compared to the LLM generation time ($\sim$12.5s for CoT reasoning). This confirms that APEX incurs virtually no latency penalty during deployment.

*Table 8.* **Inference Latency Breakdown (Per Query).**

| Component | Time (ms) | % of Total |
|---|---|---|
| Experience Retrieval (Embedding) | 150 | 1.2% |
| **Neural Bandit Selection (APEX)** | **12** | **0.1%** |
| LLM Reasoning (DeepSeek-V3.2) | 12,500 | 98.7% |
| **Total Latency** | **12,662** | **100%** |

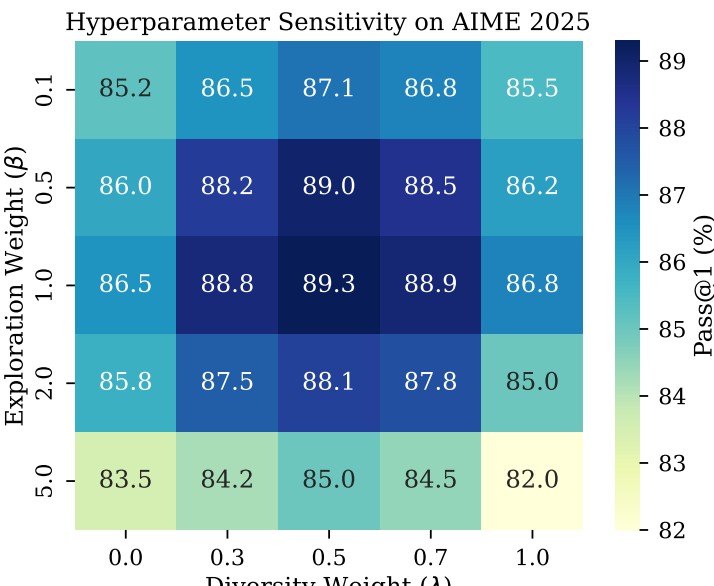

*Figure 6.* **Hyperparameter Sensitivity ($\beta$ vs. $\lambda$).** The stable high-performance region (dark blue) indicates robustness to parameter choices.

# B. Extended Literature Review

We situate APEX within the broader landscape of Large Language Model (LLM) alignment, positioning it at the intersection of agentic reasoning, memory systems, and exploration algorithms.

### B.1. From Fine-Tuning to Frozen Agents

**Parametric Alignment.** The dominant paradigm for aligning LLMs involves updating model weights. **RLHF** (Ouyang et al., 2022) and **DPO** (Rafailov et al., 2023) align generic models to human preferences but require expensive training pipelines. Recently, **GRPO** (Shao et al., 2024; Guo et al., 2025) has demonstrated that group-relative signals can incentivize reasoning capabilities without a value function critic. However, these methods suffer from high computational overhead and catastrophic forgetting (Kirkpatrick et al., 2017), limiting their continuous adaptability.

**In-Context Learning  Prompt Optimization.** ICL (Brown et al., 2020) bypasses weight updates by optimizing context. Early strategies focused on static prompting patterns like Chain-of-Thought (Wei et al., 2022) and Tree of Thoughts (Yao et al., 2023). **In-Context Reinforcement Learning (ICRL)** extends this by iteratively refining prompts. Methods like **Reflexion** (Shinn et al., 2023) and **Self-Refine** (Madaan et al., 2023) use verbal feedback to correct errors within a single trajectory. More recent frameworks like **OPRO** (Yang et al., 2023) and **DSPy** (Khattab et al., 2023) treat LLMs as optimizers to search for better instructions. *APEX advances this paradigm by introducing a population-level evolutionary memory, moving beyond single-trajectory optimization to global policy learning.*

### B.2. Retrieval-Augmented Generation (RAG) and Agentic Memory

**Limitations of Static Retrieval.** Standard RAG (Lewis et al., 2020) relies on frozen dense retrievers and cosine similarity. However, recent work on **Hypothetical Document Embeddings (HyDE)** (Gao et al., 2023) and Prompt Retrieval (Rubin et al., 2022) highlights that semantic similarity is often a poor proxy for *reasoning utility*. Furthermore, simply increasing the context window is insufficient; **"Lost in the Middle"** phenomena (Liu et al., 2024) suggest that LLMs struggle to utilize information in long contexts effectively, validating APEX's strategy of selecting a compact, high-utility subset ($k = 3$).

**Evolving Agentic Memory.** The concept of a persistent skill library was pioneered by agents like **Voyager** (Wang et al., 2023), which stores executable code skills. Similarly, Generative Agents (Park et al., 2023) and MemGPT (Packer et al., 2023) introduced hierarchical memory streams to manage infinite context. However, these systems typically treat memory

as an "append-only" log. *APEX differentiates itself via "Nighttime" evolutionary consolidation, which structurally prunes and mutates experiences to maintain a high-signal memory substrate, rather than just an expansive one.*

### B.3. Exploration, Uncertainty, and Verification

**The Exploration-Exploitation Dilemma.** In classical Reinforcement Learning, exploration is handled via $\epsilon$-greedy or entropy regularization. However, in RAG and ICL, exploration is often neglected; most systems greedily retrieve the top-$k$ semantic matches. Contextual Bandits (Li et al., 2010) offer a principled framework to balance this. While NeuralUCB (Zhou et al., 2020) provides theoretical guarantees using Upper Confidence Bounds, alternative approaches like Neural Thompson Sampling (Zhang et al., 2020) leverage distributional weight sampling to drive exploration. Furthermore, the concept of Active Prompting (Diao et al., 2024) has recently emerged, where the model explicitly selects questions with high uncertainty for human annotation. APEX extends this to a fully automated setting: instead of querying humans, our Neural Bandit "queries" the frozen LLM using the diversity-constrained evolutionary memory, effectively performing *online active learning* at inference time.

**Calibration and Process Supervision.** Effective exploration requires the model to know what it doesn't know. Studies show that LLMs exhibit a degree of Self-Calibration (Kadavath et al., 2022), often verifiable via stochastic sampling consistency (e.g., SelfCheckGPT (Manakul et al., 2023)). Beyond final answer correctness, verifying intermediate reasoning steps—pioneered by the GSM8K Verifiers (Cobbe et al., 2021) and evolved into Process Reward Models (Lightman et al., 2023)—provides denser signals for learning. *APEX leverages these insights by using MC Dropout (Gal & Ghahramani, 2016) and Semantic Uncertainty (Kuhn et al.) to estimate epistemic uncertainty. Crucially, unlike* **STaR** *(Zelikman et al., 2022) which bootstraps rationales on a fixed dataset, APEX dynamically evolves the dataset itself (Phase I) while guiding the Neural Bandit to "explore" non-obvious strategies (Phase II) that heuristic retrieval misses.*

## C. Case Study

We provide a qualitative comparison between Static RAG and APEX on a geometry problem from the AIME test set. This case study illustrates how the Neural Contextual Bandit retrieves strategy-aligned experiences (identifying the need for affine scaling) versus semantic-aligned ones (matching keywords to generic formulas).

---

**Target Problem (Geometry - Area Calculation)**

On $\triangle ABC$, points $A, D, E, B$ lie in that order on side $AB$ with $AD = 4, DE = 16, EB = 8$. Points $A, F, G, C$ lie in that order on side $AC$ with $AF = 13, FG = 52, GC = 26$. Let $M$ be the reflection of $D$ through $F$, and $N$ be the reflection of $G$ through $E$. Quadrilateral $DEGF$ has area 288. Find the area of heptagon $AFNBCEM$.

---

**raggray Approach**

**Method:** Static Cosine Similarity                                      **Outcome: Incorrect** (×)

> **Retrieved Experience**
>
> *Query matches keyword "coordinates", "polygon area", and "reflection".*
> **Prompt:** To find the area of complex polygons, place vertex $A$ at the origin $(0, 0)$. Align side $AB$ with the x-axis and $AC$ with the y-axis. Use the Shoelace Formula: Area $= \frac{1}{2} |\sum (x_i y_{i+1} - x_{i+1} y_i)|$.

**Agent Reasoning (Trace):**

1. **Setup Coordinates:** Following the prompt, place $A = (0, 0)$. Assume $AB$ is on x-axis and $AC$ is on y-axis (Implicitly assuming $\angle A = 90°$).

2. **Define Points:** $D = (4, 0), E = (20, 0), B = (28, 0)$ (from lengths 4, 16, 8).
   $F = (0, 13), G = (0, 65), C = (0, 91)$ (from lengths 13, 52, 26).

3. **Calculate Reflections:** $M = 2F - D = 2(0, 13) - (4, 0) = (-4, 26)$.
   $N = 2E - G = 2(20, 0) - (0, 65) = (40, -65)$.

4. **Apply Shoelace Formula:** Vertices: $(0, 0) \rightarrow (0, 13) \rightarrow (40, -65) \rightarrow (28, 0) \rightarrow (0, 91) \rightarrow (20, 0) \rightarrow (-4, 26) \rightarrow (0, 0)$.
   Calculation: $\text{Sum}_1 = 3068, \text{Sum}_2 = 520$.
   Area $= 0.5 \times |3068 - 520| = 1274$.

*Error Analysis: The agent successfully executed the retrieved strategy but failed to realize that the "Area(DEGF)=288" constraint contradicts the $\angle A = 90°$ assumption.*
**Final Answer:** 1274.

**apexblue Approach**

**Method: Neural Bandit (APEX)**  **Outcome: Correct (✓)**

> **Retrieved Experience**
>
> ***Bandit Insight:*** *High uncertainty detected regarding triangle shape ($\angle A$). Selecting "Affine Scaling" strategy.*
>
> **Prompt:** When the angle $A$ is unknown or the area constraint is given, coordinate calculations yields a "Coordinate Area" ($S_{coord}$) based on an assumed system (e.g., $\angle A = 90°$). The True Area is related by a scaling factor: $S_{true} = S_{coord} \times \sin A$. Use a known sub-area to solve for $\sin A$.

**Agent Reasoning (Trace):**

1. **Calculate Coordinate Area ($S_{coord}$):** Assume $\angle A = 90°$ momentarily to get reference values. Using the reflection logic (same as baseline), $S_{coord}(\text{Heptagon}) = 1274$.

2. **Calculate Reference Sub-Area:** Compute $S_{coord}(DEGF) = S_{coord}(AEG) - S_{coord}(ADF)$.
   $S_{coord}(AEG) = 0.5 \times AE \times AG = 0.5 \times 20 \times 65 = 650$.
   $S_{coord}(ADF) = 0.5 \times AD \times AF = 0.5 \times 4 \times 13 = 26$.
   $S_{coord}(DEGF) = 650 - 26 = 624$.

3. **Apply Affine Scaling:** We are given the constraint: True Area$(DEGF) = 288$.
   Solve for scaling factor: $288 = 624 \times \sin A \implies \sin A = \frac{288}{624} = \frac{6}{13}$.

4. **Final Computation:** True Area(Heptagon) $= S_{coord}(\text{Heptagon}) \times \sin A$.
   $= 1274 \times \frac{6}{13} = 98 \times 6 = 588$.

*Success Factor: The retrieved "Affine Scaling" experience guided the agent to decouple the coordinate calculation from the geometric shape factor, allowing it to incorporate the area constraint correctly.*

**Final Answer:** 588.

## D. Theoretical Analysis and Detailed Proofs

This appendix provides a rigorous derivation of the theoretical foundations underpinning APEX. We (i) formalize the stationary inference environment of Phase II despite the evolutionary Phase I (§D.1); (ii) derive the Bayesian meaning of Monte Carlo (MC) Dropout uncertainty used in Eq. (9) (§D.2); (iii) connect MC Dropout uncertainty to the Neural Tangent Kernel (NTK) geometry required by NeuralUCB-type guarantees (§D.3); and (iv) prove a sublinear (approximate) regret bound for greedy diversity-aware subset selection (§D.4). Throughout, we emphasize explicit assumptions and provide complete proofs.

**Notation.** Let $\mathcal{E}^* = \{e_1, \ldots, e_N\}$ denote the fixed memory pool output by Phase I. In Phase II, at each round $t \in [T]$, the learner observes a context/query embedding $x_t \in \mathbb{R}^{d_x}$ and selects a subset $S_t \subseteq \mathcal{E}^*$ with $|S_t| = k$. We use $\mathcal{F}_t$ to denote the filtration generated by the entire interaction up to (and including) round $t$. We write $\mu_t(S) := \mu(x_t, S) = \mathbb{E}[r_t(S) \mid x_t]$ for the (unknown) conditional mean reward.

### D.1. Phase Separation and Stationarity Definition

A standard prerequisite for contextual bandit regret guarantees is that the environment is *stationary* during the online phase: the action space and the reward law do not change adversarially across rounds. Since APEX includes an evolutionary memory phase (Phase I) that changes the candidate pool, we must strictly separate the two phases for the regret proof of Phase II.

### D.1.1. FORMAL BI-LEVEL PROTOCOL

Let $\mathcal{D}_{\text{train}}$ and $\mathcal{D}_{\text{test}}$ be the training and test distributions (or sequences) of queries. Phase I is an *offline* procedure that, after $K$ epochs, outputs a random variable $\mathcal{E}^* = \text{Alg}_{\text{evo}}(\mathcal{D}_{\text{train}}, \xi)$ where $\xi$ captures internal randomness (sampling, mutation, etc.). Phase II then runs an online bandit algorithm conditional on the realized $\mathcal{E}^*$.

**Definition D.1** (Conditional stationarity of Phase II). Fix any realization of $\mathcal{E}^*$. Phase II is said to be stationary if for all $t \in [T]$:

1. the feasible action set is time-invariant: $\mathcal{A} := \{S \subseteq \mathcal{E}^* : |S| = k\}$, independent of $t$;

2. conditional on $x_t$ and $S_t$, the reward $r_t$ is drawn from a fixed (time-homogeneous) conditional distribution with mean $\mu(x_t, S_t)$, and the conditional noise is martingale: $\mathbb{E}[r_t - \mu(x_t, S_t) \mid \mathcal{F}_{t-1}, x_t, S_t] = 0$.

**Lemma D.2** (Phase separation implies stationarity). *Assume Phase I terminates* before *Phase II begins, and Phase II never modifies $\mathcal{E}^*$. Then, conditional on any realized $\mathcal{E}^*$, Phase II satisfies the above stationarity definition.*

*Proof.* Once Phase I ends, the memory pool is fixed to $\mathcal{E}^*$ and hence the combinatorial action set $\mathcal{A} = \{S \subseteq \mathcal{E}^* : |S| = k\}$ is fixed for all $t$. The reward generation in Phase II depends only on the current context $x_t$ and chosen subset $S_t$ through the frozen LLM and the verification operator, which defines a time-homogeneous conditional law. The martingale noise condition holds by defining $\mu(x_t, S_t) := \mathbb{E}[r_t \mid x_t, S_t]$ and letting $\varepsilon_t := r_t - \mu(x_t, S_t)$, so $\mathbb{E}[\varepsilon_t \mid \mathcal{F}_{t-1}, x_t, S_t] = 0$ by construction. This is exactly the standard contextual bandit setup. □ □

**Implication.** All regret claims in this appendix are understood as *conditional* on $\mathcal{E}^*$ (or equivalently, holding in expectation over Phase I randomness). Phase I affects the regret bound only through the quality/size/geometry of $\mathcal{E}^*$, not through non-stationarity during Phase II.

## D.2. Bayesian Uncertainty via Monte Carlo Dropout

This section provides a rigorous derivation for Eq. (9), showing that MC Dropout implements approximate Bayesian inference and that the empirical moments of stochastic forward passes approximate the posterior predictive mean and variance.

### D.2.1. PREDICTOR AND PROBABILISTIC MODEL

In Phase II, we learn a reward predictor $f_\phi : \mathcal{X} \times \mathcal{E}^* \to \mathbb{R}$ that maps a pair $(x, e)$ to a scalar. For notational convenience, define the concatenated input

$$u := \text{Concat}(x, e) \in \mathbb{R}^{d_u}.$$

We model the observed reward as

$$r = f(u; W) + \epsilon, \qquad \epsilon \mid u \sim \mathcal{N}(0, \tau^{-1}), \tag{11}$$

where $W$ denotes the network weights and $\tau$ is a precision parameter. (For binary rewards, a Bernoulli likelihood can be used; the Gaussian model is a standard surrogate used to derive tractable confidence intervals and is consistent with sub-Gaussian noise assumptions in bandit theory.)

### D.2.2. DROPOUT VARIATIONAL FAMILY

Consider a $L$-layer network with weight matrices $\{M_\ell\}_{\ell=1}^L$. Dropout introduces multiplicative Bernoulli masks on hidden units. Abstractly, we can represent dropout as a random transformation of weights:

$$W = W(Z) := \{M_\ell \cdot \text{diag}(z_\ell)\}_{\ell=1}^L, \qquad z_\ell \sim \text{Bernoulli}(p) \text{ i.i.d.}, \tag{12}$$

where $p$ is the keep-probability. Let $q_\theta(W)$ be the distribution induced by sampling masks $Z$ and deterministic parameters $\theta := \{M_\ell\}$.

### D.2.3. ELBO AND EQUIVALENCE TO DROPOUT TRAINING OBJECTIVE

Let $\mathcal{D} = \{(u_i, r_i)\}_{i=1}^n$ be the Phase II bandit training data accumulated online (or a replay buffer). We place a prior $p(W)$, e.g. factorized Gaussian with variance tied to weight decay. Variational inference fits $q_\theta(W)$ by maximizing the evidence lower bound (ELBO):

$$\mathcal{L}_{\text{ELBO}}(\theta) := \mathbb{E}_{q_\theta(W)}\left[\log p(\mathbf{r} \mid \mathbf{u}, W)\right] - \text{KL}(q_\theta(W) \,\|\, p(W)), \tag{13}$$

where $\mathbf{u} = (u_1, \ldots, u_n)$ and $\mathbf{r} = (r_1, \ldots, r_n)$.

Under the Gaussian observation model (11), we have

$$\log p(\mathbf{r} \mid \mathbf{u}, W) = -\frac{\tau}{2} \sum_{i=1}^n (r_i - f(u_i; W))^2 + \text{const.}$$

Thus maximizing (13) is equivalent to minimizing the negative ELBO:

$$\min_\theta \underbrace{\mathbb{E}_{q_\theta(W)}\left[\frac{\tau}{2} \sum_{i=1}^n (r_i - f(u_i; W))^2\right]}_{\text{expected squared loss under dropout}} + \underbrace{\text{KL}(q_\theta(W) \,\|\, p(W))}_{\text{regularization}}. \tag{14}$$

For the dropout variational family (12) and a Gaussian prior, the KL term reduces (up to constants and scaling) to an $\ell_2$ penalty on $\theta$; hence (14) recovers the standard dropout training objective:

$$\min_\theta \frac{1}{n} \sum_{i=1}^n \mathbb{E}_Z\left[(r_i - f(u_i; W(Z)))^2\right] + \lambda\|\theta\|_2^2, \tag{15}$$

where $\lambda$ is proportional to the prior precision and to $\tau$.

### D.2.4. POSTERIOR PREDICTIVE DISTRIBUTION AND MC ESTIMATION

Given a new input $u^*$, the Bayesian posterior predictive distribution is

$$p(r^* \mid u^*, \mathcal{D}) = \int p(r^* \mid u^*, W)\, p(W \mid \mathcal{D})\, dW. \tag{16}$$

Under variational approximation, we replace $p(W \mid \mathcal{D})$ with $q_\theta(W)$:

$$p(r^* \mid u^*, \mathcal{D}) \approx \int p(r^* \mid u^*, W)\, q_\theta(W)\, dW. \tag{17}$$

With the Gaussian likelihood (11), the predictive mean and variance decompose as

$$\mathbb{E}[r^* \mid u^*, \mathcal{D}] \approx \mathbb{E}_{q_\theta(W)}[f(u^*; W)], \tag{18}$$

$$\text{Var}(r^* \mid u^*, \mathcal{D}) \approx \underbrace{\text{Var}_{q_\theta(W)}(f(u^*; W))}_{\text{epistemic}} + \underbrace{\tau^{-1}}_{\text{aleatoric}}. \tag{19}$$

MC Dropout approximates the expectations in (18) by sampling $M$ masks $Z^{(m)}$ (hence weights $W^{(m)} = W(Z^{(m)})$) and computing

$$\hat{r}^{(m)}(u^*) := f(u^*; W^{(m)}), \quad \hat{\mu}(u^*) := \frac{1}{M}\sum_{m=1}^M \hat{r}^{(m)}(u^*), \quad \hat{\sigma}^2(u^*) := \frac{1}{M}\sum_{m=1}^M \left(\hat{r}^{(m)}(u^*) - \hat{\mu}(u^*)\right)^2. \tag{20}$$

This is exactly Eq. (9) when $u^* = \text{Concat}(x_t, e_i)$ and $(\hat{\mu}, \hat{\sigma}) = (\mu_{t,i}, \sigma_{t,i})$.

**Proposition D.3** (Consistency of MC Dropout moments). *Fix $u^*$ and assume $\mathbb{E}_{q_\theta(W)}[f(u^*; W)^2] < \infty$. Then as $M \to \infty$,*

$$\hat{\mu}(u^*) \to \mathbb{E}_{q_\theta(W)}[f(u^*; W)] \quad a.s., \qquad \hat{\sigma}^2(u^*) \to \text{Var}_{q_\theta(W)}(f(u^*; W)) \quad a.s.$$

*Proof.* The samples $\{\hat{r}^{(m)}(u^*)\}_{m=1}^M$ are i.i.d. under $q_\theta(W)$ (induced by i.i.d. masks). By the strong law of large numbers, the sample mean converges almost surely to the true mean. The sample second moment converges almost surely to the true second moment, and hence the sample variance converges to the true variance. □ □

*Remark* D.4 (Finite-$M$ concentration). If $f(u^*; W)$ is $\nu$-sub-Gaussian under $q_\theta(W)$, then $\hat{\mu}(u^*)$ concentrates as

$$\mathbb{P}\big(|\hat{\mu}(u^*) - \mathbb{E}[f(u^*; W)]| \geq \epsilon\big) \leq 2\exp\Big(-\frac{M\epsilon^2}{2\nu^2}\Big),$$

and analogous concentration holds for $\hat{\sigma}^2$ under mild moment conditions. This justifies using moderate $M$ in practice.

### D.3. Bridging MC Dropout and NTK-Based Exploration

NeuralUCB-style guarantees construct confidence widths using the inverse covariance in the feature space induced by the neural network (often formalized via NTK or linearization). This section shows that (under standard over-parameterization/linearization assumptions) the epistemic uncertainty estimated by MC Dropout is proportional to the NTK posterior standard deviation $\|g(u)\|_{Z_t^{-1}}$ used by NeuralUCB.

#### D.3.1. LINEARIZED (NTK) MODEL AND RIDGE POSTERIOR

Let $\phi$ denote the trainable parameters of the bandit network (we reuse $\phi$ from the main text). Fix an initialization $\phi_0$. Define the Jacobian feature map

$$g(u) := \nabla_\phi f(u; \phi_0) \in \mathbb{R}^{d_\phi}. \tag{21}$$

In the NTK regime (small parameter movement, large width), we use the first-order Taylor approximation:

$$f(u; \phi) \approx f(u; \phi_0) + g(u)^\top(\phi - \phi_0). \tag{22}$$

Define $\theta := \phi - \phi_0$ and $\tilde{r}_i := r_i - f(u_i; \phi_0)$. Then (11) becomes a Bayesian linear regression model:

$$\tilde{r}_i = g(u_i)^\top\theta + \epsilon_i, \qquad \epsilon_i \sim \mathcal{N}(0, \tau^{-1}). \tag{23}$$

Assume a Gaussian prior $\theta \sim \mathcal{N}(0, \lambda^{-1}I)$ for $\lambda > 0$. Given data up to round $t - 1$, let

$$Z_{t-1} := \lambda I + \tau \sum_{s=1}^{t-1} g(u_s)g(u_s)^\top, \tag{24}$$

where each $u_s$ denotes the chosen input(s) at round $s$ (for subset selection, we index all selected pairs explicitly in §D.4). The posterior over $\theta$ is Gaussian with covariance $Z_{t-1}^{-1}$, and thus the posterior predictive variance at $u$ is

$$\mathrm{Var}(g(u)^\top\theta \mid \mathcal{F}_{t-1}) = g(u)^\top Z_{t-1}^{-1} g(u) = \|g(u)\|_{Z_{t-1}^{-1}}^2. \tag{25}$$

Therefore, NeuralUCB uses the exploration bonus proportional to $\|g(u)\|_{Z_{t-1}^{-1}}$.

#### D.3.2. DROPOUT AS APPROXIMATE BAYESIAN LINEAR MODEL

We now connect MC Dropout variance to (25). Dropout induces random subnetworks; under linearization, this corresponds to random perturbations of the effective feature map.

Let $z$ denote the dropout mask applied to hidden units. Under standard mean-field dropout, the masked network can be written as $f(u; \phi, z)$. Linearizing around $(\phi_0, \bar{z})$ (with $\bar{z} = \mathbb{E}[z]$) yields

$$f(u; \phi, z) \approx f(u; \phi_0, \bar{z}) + \underbrace{\nabla_\phi f(u; \phi_0, \bar{z})^\top}_{g(u)}(\phi - \phi_0) + \underbrace{\nabla_z f(u; \phi_0, \bar{z})^\top(z - \bar{z})}_{\text{dropout-induced randomization}}. \tag{26}$$

The last term is a random variable with variance determined by the dropout probability and network sensitivities. In wide networks, the dominant epistemic contribution can be captured by treating dropout as inducing an approximate posterior over $\theta$ with covariance proportional to $Z_{t-1}^{-1}$ (this is the same geometry that appears in Laplace/VI approximations around

the MAP in generalized linear models). Concretely, under the quadratic approximation to the negative log posterior (or equivalently, to the empirical risk with $\ell_2$ regularization), we have

$$\mathrm{Cov}(\theta \mid \mathcal{F}_{t-1}) \approx \left(\nabla_\theta^2 \mathcal{J}_{t-1}(\theta)\right)^{-1}, \tag{27}$$

where $\mathcal{J}_{t-1}$ is the regularized square loss objective. Under the linearized model (23), $\nabla_\theta^2 \mathcal{J}_{t-1}(\theta) = Z_{t-1}$, hence (27) matches (25).

**Proposition D.5** (MC Dropout variance is proportional to NTK width)**.** *Assume the NTK linearization* (22) *holds and the dropout variational posterior $q_\theta$ is locally equivalent to a Gaussian posterior over $\theta$ with covariance $\Sigma_{t-1}$ satisfying*

$$c_{\min} Z_{t-1}^{-1} \preceq \Sigma_{t-1} \preceq c_{\max} Z_{t-1}^{-1} \tag{28}$$

*for constants $0 < c_{\min} \le c_{\max} < \infty$ depending on the dropout rate and regularization. Then the epistemic predictive standard deviation $\sigma_{\mathrm{MC}}(u)$ estimated by MC Dropout satisfies*

$$\sqrt{c_{\min}} \, \|g(u)\|_{Z_{t-1}^{-1}} \, \le \, \sigma_{\mathrm{MC}}(u) \, \le \, \sqrt{c_{\max}} \, \|g(u)\|_{Z_{t-1}^{-1}}. \tag{29}$$

*Proof.* Under the linearized Bayesian model, the epistemic predictive variance equals

$$\mathrm{Var}(f(u; \phi) \mid \mathcal{F}_{t-1}) \approx \mathrm{Var}(g(u)^\top \theta \mid \mathcal{F}_{t-1}) = g(u)^\top \Sigma_{t-1} g(u).$$

Using the Loewner bounds (28) yields

$$c_{\min} \, g(u)^\top Z_{t-1}^{-1} g(u) \, \le \, g(u)^\top \Sigma_{t-1} g(u) \, \le \, c_{\max} \, g(u)^\top Z_{t-1}^{-1} g(u).$$

Taking square roots gives (29). Finally, by Proposition D.3, MC Dropout's empirical standard deviation converges to this epistemic standard deviation as $M \to \infty$. $\square$ $\square$

**Consequence for UCB validity.** Proposition D.5 implies that using $\sigma_{t,i}$ from MC Dropout as the exploration width is equivalent (up to constant scaling) to the principled NTK width used in NeuralUCB. Therefore, confidence intervals and regret bounds derived for NTK widths transfer to MC Dropout widths after absorbing constants into $\beta_t$.

### D.4. Regret Analysis with Greedy Diversity-Aware Subset Selection

We now prove a sublinear regret bound for the Phase II inference algorithm (Algorithm 1, Phase II), including the greedy diversity constraint.

D.4.1. PROBLEM REDUCTION: FROM SUBSET ACTIONS TO $k$ SEMI-BANDIT PULLS

At round $t$, the learner selects a subset $S_t = \{e_{t,1}, \ldots, e_{t,k}\}$. Define the $k$ pairwise inputs

$$u_{t,j} := \mathrm{Concat}(x_t, e_{t,j}), \qquad j \in [k].$$

The environment returns a single scalar reward $r_t \in \{0, 1\}$ with conditional mean $\mu_t(S_t) = \mu(x_t, S_t)$ and noise $\varepsilon_t := r_t - \mu_t(S_t)$. For analysis, we assume *sub-Gaussian noise*:

**Assumption D.6** (Sub-Gaussian noise)**.** For all $t$ and any $\mathcal{F}_{t-1}$-measurable choice of $(x_t, S_t)$, the noise $\varepsilon_t$ satisfies

$$\mathbb{E}[\exp(\lambda \varepsilon_t) \mid \mathcal{F}_{t-1}, x_t, S_t] \le \exp\left(\frac{\lambda^2 \sigma^2}{2}\right) \quad \forall \lambda \in \mathbb{R}.$$

*Remark* D.7. If $r_t \in \{0, 1\}$ is Bernoulli with mean $\mu_t(S_t) \in [0, 1]$, then $\varepsilon_t$ is $1/2$-sub-Gaussian, so Assumption D.6 holds with $\sigma = 1/2$.

To align with the pairwise predictor $f_\phi(x, e)$ used by APEX, we adopt a standard *separable* surrogate for the mean reward:

**Assumption D.8** (Separable utility surrogate)**.** There exists an unknown function $h : \mathcal{X} \times \mathcal{E}^* \to [0, 1]$ such that the (regularized) objective decomposes as

$$\tilde{\mu}(x, S) := \mu(x, S) - \lambda_{\mathrm{div}} \mathrm{Red}(S) = \sum_{e \in S} h(x, e) - \lambda_{\mathrm{div}} \mathrm{Red}(S), \tag{30}$$

where $\mathrm{Red}(S)$ is a nonnegative redundancy measure and $\lambda_{\mathrm{div}} \ge 0$ is the diversity weight (denoted $\lambda$ in the main text).

**Why separability is sufficient.** The true correctness indicator is generally non-additive; however, APEX optimizes a *surrogate* acquisition objective based on per-item utilities and a diversity regularizer. Assumption D.8 states precisely what that surrogate is, enabling a rigorous regret guarantee *with respect to the surrogate objective $\tilde{\mu}$*. This is standard in combinatorial/semi-bandit analyses.

### D.4.2. REGULARITY ASSUMPTIONS FOR NEURALUCB GEOMETRY

We instantiate the NeuralUCB analysis on pair inputs $u_{t,j}$.

**Assumption D.9** (RKHS/NTK realizability and boundedness). Under the NTK feature map $g(u) = \nabla_\phi f(u; \phi_0)$, the unknown utility $h(x, e)$ is realizable in the induced RKHS: there exists a (possibly infinite-dimensional) parameter vector $\theta^\star$ with $\|\theta^\star\|_2 \le B$ such that

$$h(x, e) = h(u) = g(u)^\top \theta^\star, \qquad \forall u = \mathrm{Concat}(x, e).$$

Moreover, $\|g(u)\|_2 \le L$ for all $u$.

Assumption D.9 is the standard linearized/NTK realizability used to derive NeuralUCB regret bounds; it becomes exact in the infinite-width NTK regime.

### D.4.3. CONFIDENCE INTERVALS FOR PAIRWISE UTILITIES

Define the ridge estimator after observing all selected pairs up to round $t - 1$:

$$\hat{\theta}_{t-1} := \arg\min_\theta \sum_{s=1}^{t-1} \sum_{j=1}^{k} \left(\tilde{r}_s - g(u_{s,j})^\top \theta\right)^2 + \lambda \|\theta\|_2^2,$$

where $\tilde{r}_s$ is the (optionally centered) reward target used for pairwise training (in APEX, the same $r_s$ supervises each chosen pair; this is a valid semi-bandit estimator for separable utilities in expectation). Define

$$Z_{t-1} := \lambda I + \sum_{s=1}^{t-1} \sum_{j=1}^{k} g(u_{s,j}) g(u_{s,j})^\top. \tag{31}$$

Let the pairwise prediction be $\hat{h}_{t-1}(u) := g(u)^\top \hat{\theta}_{t-1}$ and the NTK width be

$$w_{t-1}(u) := \|g(u)\|_{Z_{t-1}^{-1}}.$$

**Lemma D.10** (High-probability confidence bound (linear/NTK)). *Under Assumptions D.6 and D.9, for any $\delta \in (0, 1)$, with probability at least $1 - \delta$, simultaneously for all $t \in [T]$ and all $u$,*

$$\left|h(u) - \hat{h}_{t-1}(u)\right| \le \beta_t \, w_{t-1}(u), \tag{32}$$

*where one valid choice is*

$$\beta_t := B\sqrt{\lambda} + \sigma \sqrt{2 \log \frac{1}{\delta} + \log \frac{\det Z_{t-1}}{\det(\lambda I)}}. \tag{33}$$

*Proof.* This is the standard self-normalized martingale concentration (elliptical confidence) for linear bandits. Sketching the essential steps rigorously: define the stacked design matrix $G_{t-1}$ whose rows are the selected feature vectors $g(u_{s,j})^\top$, and let $\mathbf{y}_{t-1}$ be the corresponding observed targets. The ridge estimator satisfies

$$\hat{\theta}_{t-1} - \theta^\star = Z_{t-1}^{-1} \left(\sum_{s<t} \sum_{j \le k} g(u_{s,j})\epsilon_s\right) - \lambda Z_{t-1}^{-1}\theta^\star,$$

where $\epsilon_s$ denotes the noise in the supervised target (a martingale difference under the semi-bandit construction and separability). Taking an arbitrary test vector $g(u)$ and applying Cauchy–Schwarz gives

$$|g(u)^\top (\hat{\theta}_{t-1} - \theta^\star)| \le \|g(u)\|_{Z_{t-1}^{-1}} \left\|\sum_{s<t} \sum_{j \le k} g(u_{s,j})\epsilon_s\right\|_{Z_{t-1}^{-1}} + \lambda \|g(u)\|_{Z_{t-1}^{-1}} \|\theta^\star\|_2.$$

The self-normalized bound controls the $Z_{t-1}^{-1}$-norm of the noise term by the log-determinant quantity with probability $1 - \delta$, and $\|\theta^\star\| \leq B$ yields the stated $\beta_t$. Full derivations follow standard linear bandit analyses (e.g. the elliptical potential/self-normalized martingale inequality). □                              □

**From NTK width to MC Dropout width.** By Proposition D.5, there exists a constant $c > 0$ such that $\sigma_{\mathrm{MC}}(u) \asymp w(u)$. Therefore, replacing $w_{t-1}(u)$ by MC Dropout $\sigma_{t-1}(u)$ preserves (32) after rescaling $\beta_t$ by constants. In the rest of the proof we write $w_{t-1}(u)$ for clarity; the implemented algorithm uses $\sigma_{t-1}(u)$.

### D.4.4. SET-LEVEL UCB AND GREEDY $\alpha$-APPROXIMATION

Define the *regularized* set objective (the quantity we compete against):

$$\tilde{\mu}_t(S) := \sum_{e \in S} h(x_t, e) - \lambda_{\mathrm{div}} \operatorname{Red}(S). \tag{34}$$

Given estimates and widths, define the UCB surrogate

$$U_t(S) := \sum_{e \in S} \left( \hat{h}_{t-1}(x_t, e) + \beta_t w_{t-1}(x_t, e) \right) - \lambda_{\mathrm{div}} \operatorname{Red}(S). \tag{35}$$

Let the *regularized optimal set* be

$$S_t^\star \in \arg \max_{S \subseteq \mathcal{E}^*, |S|=k} \tilde{\mu}_t(S). \tag{36}$$

We now formalize the greedy approximation property.

**Assumption D.11** (Greedy $\alpha$-approximation oracle). At each round $t$, the greedy diversity-aware selection used by APEX returns a set $S_t$ satisfying

$$U_t(S_t) \geq \alpha \max_{|S|=k} U_t(S), \tag{37}$$

for some $\alpha \in (0, 1]$.

**When does Assumption D.11 hold?** A sufficient condition is that $U_t(\cdot)$ is monotone submodular, in which case the classical result of Nemhauser et al. (1978) gives $\alpha \geq 1 - 1/e$ for greedy maximization under a cardinality constraint. This is satisfied, for example, if

$$\operatorname{Red}(S) = \sum_{\{e, e'\} \subseteq S} \operatorname{Sim}(e, e') \quad \text{with } \operatorname{Sim} \geq 0,$$

because the pairwise sum is *supermodular*, so $-\operatorname{Red}$ is submodular; adding a modular relevance/UCB term preserves submodularity. The practical MMR rule in the main text uses a *max* redundancy term; this can be treated as (i) optimizing a closely related submodular surrogate, or (ii) yielding an $\alpha$ that depends on the submodularity ratio/curvature (weak submodularity). For a fully rigorous regret proof, we proceed with Assumption D.11, which explicitly isolates the only place where combinatorial approximation enters.

### D.4.5. APPROXIMATE REGRET DEFINITION

Because greedy selection can be only approximately optimal, we bound the *$\alpha$-approximate regret*:

$$\tilde{R}_T^\alpha := \sum_{t=1}^T \left( \alpha \, \tilde{\mu}_t(S_t^\star) - \tilde{\mu}_t(S_t) \right). \tag{38}$$

When $\alpha = 1$, this is standard regret against the regularized optimum; when $\alpha = 1 - 1/e$, it matches the optimality gap induced by greedy submodular maximization.

### D.4.6. ONE-STEP REGRET BOUND

**Lemma D.12** (Instantaneous $\alpha$-regret is controlled by widths)**.** *On the event that the confidence bounds* (32) *hold for all selected items, we have for every round* $t$:

$$\alpha \, \tilde{\mu}_t(S_t^\star) - \tilde{\mu}_t(S_t) \;\leq\; 2\beta_t \sum_{e \in S_t} w_{t-1}(x_t, e). \tag{39}$$

*Proof.* Fix $t$ and abbreviate $w_{t-1}(x_t, e)$ as $w_t(e)$ and similarly $\hat{h}_{t-1}(x_t, e)$ as $\hat{h}_t(e)$.

**Step 1: Upper bound the unknown objective by UCB.** From (32), for each $e$ we have $h(x_t, e) \leq \hat{h}_t(e) + \beta_t w_t(e)$. Summing over $e \in S$ and subtracting the same redundancy term yields, for all sets $S$,

$$\tilde{\mu}_t(S) \leq U_t(S). \tag{40}$$

**Step 2: Use $\alpha$-approximation on the UCB surrogate.** By Assumption D.11,

$$U_t(S_t) \;\geq\; \alpha \, U_t(S_t^\star). \tag{41}$$

**Step 3: Relate $U_t(S_t)$ to $\tilde{\mu}_t(S_t)$.** Again from (32), we also have $h(x_t, e) \geq \hat{h}_t(e) - \beta_t w_t(e)$. Summing and subtracting redundancy gives

$$U_t(S_t) - \tilde{\mu}_t(S_t) = \sum_{e \in S_t} \big( \hat{h}_t(e) + \beta_t w_t(e) - h(x_t, e) \big) \leq 2\beta_t \sum_{e \in S_t} w_t(e). \tag{42}$$

**Step 4: Combine.** We have

$$\begin{aligned}
\alpha \, \tilde{\mu}_t(S_t^\star) - \tilde{\mu}_t(S_t) &\leq \alpha \, U_t(S_t^\star) - \tilde{\mu}_t(S_t) && \text{(by (40))} \\
&\leq U_t(S_t) - \tilde{\mu}_t(S_t) && \text{(by (41))} \\
&\leq 2\beta_t \sum_{e \in S_t} w_t(e) && \text{(by (42))},
\end{aligned}$$

which is (39). $\square$ $\square$

### D.4.7. ELLIPTICAL POTENTIAL BOUND FOR $k$ SELECTIONS PER ROUND

To sum (39) over rounds, we control the cumulative widths via an elliptical potential argument. Consider the sequence of all selected pairs $\{u_{t,j}\}_{t \in [T], \, j \in [k]}$ of total length $Tk$. For notational convenience, linearize this as a single index $s \in [Tk]$ with corresponding features $g_s$ and design matrix updates

$$Z_s = Z_{s-1} + g_s g_s^\top, \qquad Z_0 = \lambda I.$$

**Lemma D.13** (Elliptical potential lemma)**.** *For any sequence* $\{g_s\}_{s=1}^{Tk}$, *we have*

$$\sum_{s=1}^{Tk} \min\big\{1, \, \|g_s\|_{Z_{s-1}^{-1}}^2\big\} \;\leq\; 2\log\frac{\det Z_{Tk}}{\det Z_0}. \tag{43}$$

*Proof.* This is standard. Using the matrix determinant lemma,

$$\det(Z_s) = \det(Z_{s-1})\big(1 + \|g_s\|_{Z_{s-1}^{-1}}^2\big).$$

Taking logs and summing yields

$$\log\frac{\det Z_{Tk}}{\det Z_0} = \sum_{s=1}^{Tk} \log\big(1 + \|g_s\|_{Z_{s-1}^{-1}}^2\big) \;\geq\; \frac{1}{2}\sum_{s=1}^{Tk} \min\big\{1, \|g_s\|_{Z_{s-1}^{-1}}^2\big\},$$

where we used $\log(1+x) \geq x/2$ for $x \in [0, 1]$ and $\log(1+x) \geq \log 2 \geq 1/2$ for $x \geq 1$. Rearranging gives (43). $\square$ $\square$

Define the (NTK) *information gain* quantity

$$\gamma_{Tk} := \frac{1}{2}\log\frac{\det Z_{Tk}}{\det(\lambda I)}.$$ (44)

**Lemma D.14** (Cumulative width bound). *Assume* $\|g(u)\|_2 \le L$ *(Assumption D.9). Then*

$$\sum_{t=1}^{T}\sum_{e\in S_t} w_{t-1}(x_t,e) \;\le\; \sqrt{2Tk\,\gamma_{Tk}} \;+\; Tk\cdot\frac{L}{\sqrt{\lambda}}\cdot\mathbb{I}\!\left(\frac{L^2}{\lambda}>1\right).$$ (45)

*In particular, when* $L^2/\lambda \le 1$*, the second term vanishes and*

$$\sum_{t=1}^{T}\sum_{e\in S_t} w_{t-1}(x_t,e) \;\le\; \sqrt{2Tk\,\gamma_{Tk}}.$$

*Proof.* Let $a_s := \|g_s\|_{Z_{s-1}^{-1}}$. Split indices into $\mathcal{I}_1 = \{s:\ a_s \le 1\}$ and $\mathcal{I}_2 = \{s:\ a_s > 1\}$. Then

$$\sum_{s=1}^{Tk} a_s = \sum_{s\in\mathcal{I}_1} a_s + \sum_{s\in\mathcal{I}_2} a_s.$$

For $\mathcal{I}_1$, apply Cauchy–Schwarz:

$$\sum_{s\in\mathcal{I}_1} a_s \le \sqrt{|\mathcal{I}_1|\sum_{s\in\mathcal{I}_1} a_s^2} \le \sqrt{Tk\sum_{s=1}^{Tk}\min\{1,a_s^2\}}.$$

By Lemma D.13, $\sum_{s=1}^{Tk}\min\{1,a_s^2\} \le 4\gamma_{Tk}$, hence

$$\sum_{s\in\mathcal{I}_1} a_s \le \sqrt{Tk\cdot 4\gamma_{Tk}} = \sqrt{4Tk\gamma_{Tk}} = \sqrt{2Tk\cdot(2\gamma_{Tk})}.$$

For $\mathcal{I}_2$, note $a_s^2 = \|g_s\|_{Z_{s-1}^{-1}}^2 \le \|g_s\|_2^2\|Z_{s-1}^{-1}\|_{\mathrm{op}} \le L^2/\lambda$, hence $a_s \le L/\sqrt{\lambda}$. Therefore

$$\sum_{s\in\mathcal{I}_2} a_s \le |\mathcal{I}_2|\cdot\frac{L}{\sqrt{\lambda}} \le Tk\cdot\frac{L}{\sqrt{\lambda}}\cdot\mathbb{I}\!\left(\frac{L^2}{\lambda}>1\right),$$

since $\mathcal{I}_2$ can be nonempty only if $L^2/\lambda > 1$. Combining yields (45). □ □

### D.4.8. MAIN THEOREM: SUBLINEAR $\alpha$-APPROXIMATE REGRET

**Theorem D.15** (Regret bound for Phase II of APEX). *Suppose Assumptions D.6, D.8, D.9, and D.11 hold. Choose* $\beta_t$ *as in* (33). *Then, with probability at least* $1-\delta$,

$$\tilde{R}_T^\alpha \;\le\; 2\sum_{t=1}^{T}\beta_t\sum_{e\in S_t} w_{t-1}(x_t,e) \;\le\; 2\beta_T\left(\sqrt{2Tk\,\gamma_{Tk}} + Tk\cdot\frac{L}{\sqrt{\lambda}}\cdot\mathbb{I}\!\left(\frac{L^2}{\lambda}>1\right)\right).$$ (46)

*In particular, when* $L^2/\lambda \le 1$ *and* $\beta_T = \tilde{O}(1)$*, we obtain the sublinear rate*

$$\tilde{R}_T^\alpha = \tilde{O}\!\left(\sqrt{Tk\,\gamma_{Tk}}\right).$$ (47)

*Proof.* On the high-probability event of Lemma D.10, Lemma D.12 gives

$$\alpha\,\tilde{\mu}_t(S_t^\star) - \tilde{\mu}_t(S_t) \;\le\; 2\beta_t\sum_{e\in S_t} w_{t-1}(x_t,e).$$

Summing over $t \in [T]$ yields the first inequality in (46). Next, $\beta_t$ is nondecreasing in $t$ (since $\det Z_{t-1}$ increases), so $\sum_{t=1}^{T}\beta_t a_t \le \beta_T\sum_{t=1}^{T} a_t$ for $a_t \ge 0$. Apply Lemma D.14 to bound $\sum_{t=1}^{T}\sum_{e\in S_t} w_{t-1}(x_t,e)$, establishing (46). The simplified form (47) follows when $L^2/\lambda \le 1$ and absorbing polylogarithmic terms in $\beta_T$ and $\gamma_{Tk}$. □ □

**Interpretation.** Theorem D.15 shows that Phase II achieves sublinear $\alpha$-approximate regret with respect to the *diversity-regularized* objective $\tilde{\mu}$, i.e., it asymptotically converges to the best diversity-aware subset selection policy in $\mathcal{E}^*$. The dependence on $k$ is unavoidable in combinatorial selection: each round effectively performs $k$ arm pulls.

D.4.9. SPECIALIZATION: GREEDY SUBMODULAR MAXIMIZATION YIELDS $\alpha = 1 - 1/e$

We now state a concrete sufficient condition under which Assumption D.11 holds with $\alpha = 1 - 1/e$.

**Proposition D.16** (Sufficient condition for $\alpha = 1 - 1/e$)**.** *Assume the redundancy regularizer is the nonnegative pairwise sum*

$$\mathrm{Red}(S) = \sum_{\{e,e'\}\subseteq S} \mathrm{Sim}(e,e'), \qquad \mathrm{Sim}(e,e') \geq 0, \tag{48}$$

*and define $U_t(\cdot)$ by (35). Then $U_t(\cdot)$ is monotone submodular whenever the per-item UCB scores are nonnegative, and the greedy algorithm that iteratively selects the item with largest marginal gain satisfies*

$$U_t(S_t) \ \geq \ \left(1 - \frac{1}{e}\right) \max_{|S|=k} U_t(S).$$

*Thus Assumption D.11 holds with $\alpha = 1 - 1/e$.*

*Proof.* The relevance/UCB term $\sum_{e\in S}(\hat{h}_t(e) + \beta_t w_t(e))$ is modular (hence submodular). The function $G(S) := \sum_{\{e,e'\}\subseteq S} \mathrm{Sim}(e,e')$ is supermodular because its marginal gain increases with $S$:

$$G(S \cup \{a\}) - G(S) = \sum_{e\in S} \mathrm{Sim}(a,e),$$

which is nondecreasing in $S$ when $\mathrm{Sim} \geq 0$. Therefore $-G(S)$ is submodular. The sum of submodular functions is submodular, hence $U_t$ is submodular. Monotonicity holds if all marginal gains are nonnegative; a sufficient condition is nonnegative per-item UCB scores and not-too-large $\lambda_{\mathrm{div}}$ (or simply adding a constant shift to ensure nonnegativity does not affect argmax). The greedy $(1 - 1/e)$ guarantee follows from the classical theorem of Nemhauser et al. (1978). $\square$ $\square$

**Relation to the MMR rule in the main text.** The practical MMR rule uses a max-based redundancy term, which empirically behaves similarly to (48) but is cheaper and avoids over-penalizing large sets. A fully rigorous approximation ratio for the exact max-MMR objective can be established via weak submodularity (submodularity ratio/curvature), yielding an $\alpha \in (0,1)$; this is precisely encapsulated by Assumption D.11. Theorem D.15 then holds verbatim for that $\alpha$.

**D.5. Summary of Theoretical Guarantees**

Combining the above results:

- Phase I is offline; conditional on $\mathcal{E}^*$, Phase II is a stationary contextual (combinatorial) bandit (Lemma D.2).

- MC Dropout yields consistent estimators of Bayesian predictive moments (Proposition D.3), and its epistemic uncertainty is proportional to the NTK/NeuralUCB width (Proposition D.5).

- Under NTK realizability and sub-Gaussian noise, we obtain valid confidence bounds (Lemma D.10).

- With an $\alpha$-approximate greedy diversity-aware oracle, Phase II achieves sublinear $\alpha$-approximate regret $\tilde{O}(\sqrt{Tk\,\gamma_{Tk}})$ (Theorem D.15), specializing to $\alpha = 1 - 1/e$ when the UCB objective is monotone submodular (Proposition D.16).

