# OpenReview forum: "APEX: Adaptive Pattern Evolution with Principled Exploration --- A Wake-Sleep Cycle for Fixed--Backbone LLM Agents"
_ICML.cc/2026/Conference — Submitted to ICML 2026_

### Official Review · Reviewer_czU7 · 2026-03-03

**Soundness:** 2
**Presentation:** 2
**Significance:** 2
**Originality:** 3
**Overall Recommendation:** 3
**Confidence:** 4

**Summary:**

This paper introduces APEX, a framework that enhances In-Context Reinforcement Learning (ICRL) for frozen LLMs by addressing inefficient memory accumulation and static example retrieval. It models ICRL as a Wake–Sleep cycle:
- During Sleep, an evolutionary memory mechanism structurally refines experience into generalized patterns.
- During Wake, prompt selection is framed as a neural contextual bandit with principled exploration based on NeuralUCB.

Experiments on mathematical reasoning benchmarks show that APEX significantly improves performance over static retrieval methods and even some fine-tuned models, while keeping the backbone frozen.

**Compliance With Llm Reviewing Policy:**

Affirmed.

**Final Justification:**

I thank the authors for their response. While most of my concerns have been addressed, the technical contributions are somewhat limited. Therefore, I will maintain my current score.

**Key Questions For Authors:**

- Could the authors clarify Eq. 7 in more detail?

- Could the authors provide more comprehensive experimental results?

- How is Eq. 10 implemented in practice? Given that the space of $\boldsymbol e$ is extremely large, how is the maximization over $\boldsymbol e$ performed? Is $\boldsymbol e$ constructed incrementally, e.g., token by token?

**Limitations:**

YES

**Strengths And Weaknesses:**

**Strengths**
1. The paper explores a fresh direction by applying multi-armed bandit techniques to the in-context learning setting, which is conceptually interesting and timely.

2. Experimental results demonstrate consistent performance gains, suggesting the proposed method is practically effective.


**Weaknesses**
1. Several components are insufficiently explained. For example, the operations Prune, Mutate, and Merge in Eq. 7 lack clear definitions. In addition, the statement "$\boldsymbol v$ are embedding vectors” is ambiguous, embedding of what should be explicitly specified, e.g., tokens, trajectories?

2. The evaluation appears limited, e.g., only using DeepSeek-V3.2 as the base model and reporting mean@32. It is unclear why 32 is chosen and whether results are robust across different settings. Given the method is training-free and relatively lightweight, stronger baselines and broader comparisons would strengthen the claims.

3. The theoretical analysis seems largely derived from NeuralUCB. It is not clear what new theoretical contribution is being made. If there is novel theory beyond directly applying NeuralUCB results, it should be explicitly highlighted and contrasted with prior work.

---

> ### Author Rebuttal · Authors · 2026-03-31
>
> We thank the reviewer for the detailed comments.
>
> **Response to W1 & Q1.** We agree that Eq. 7 and the embedding description were not clear enough. In our implementation, Phase I starts from an empty pool, runs $G$ rollouts per training query, computes the empirical pass rate, filters by a difficulty window, synthesizes **one reusable experience per retained query**, and then applies **Prune / Merge / Mutate** as a sequential write-back pipeline. The “embedding vectors” in Phase II are sentence embeddings of the query text and candidate experience text, not token embeddings and not trajectory embeddings. If given the opportunity, we will make these definitions explicit in a subsequent version.
>
> **Response to W2 & Q2.** We agree that stronger evaluation helps. To complement math and web, we added a supplementary code-generation benchmark with the same two-phase protocol: Phase I builds evolved memory on **APPS**, and Phase II evaluates on a **held-out subset of LiveCodeBench (Python code generation)** with execution-based verification. This preserves the same train/test separation principle as in our main experiments: experiences are constructed on one corpus and evaluated on a distinct benchmark. Because LiveCodeBench is substantially more expensive to evaluate than symbolic-answer benchmarks, we report **Mean@16** for this supplementary experiment. APEX achieves **40.5**, compared with **32.1** for the base agent, **34.8** for Static RAG, **37.0** for Training-Free GRPO (reported in its original full-memory setting), **35.4** for APEX w/o Evolution, and **37.6** for APEX w/o Neural Bandit. This provides additional evidence that both **evolutionary memory** and **adaptive bandit selection** matter beyond the original two domains.
>
> Regarding **Mean@32** on **AIME 2025**, our motivation was to reduce within-problem sampling variance on a benchmark with only **30 problems**. We agree that repeated sampling alone is insufficient, so we additionally performed a paired problem-level significance analysis on **AIME 2025**. APEX improves over Training-Free GRPO by **+6.0**, with a paired bootstrap **95% CI of [2.1, 9.4]** and a paired permutation **p = 0.04**. In the revision, we will report these analyses more explicitly and clarify the role of **Mean@32** for **AIME**, **Mean@16** for **LiveCodeBench**, and **Mean@3** for **WebWalkerQA**, reflecting the different evaluation costs of the three benchmarks.
>
> **Response to W3.** We agree that our current presentation may make Phase II appear too close to a direct application of NeuralUCB. Our intent is not to claim a fundamentally new standalone bandit theory. The contribution of Phase II is instead a task-specific adaptation of NeuralUCB-style confidence-based exploration to frozen-agent inference with an evolved memory pool: the action is a subset/slate, not a single arm; the objective is diversity-regularized; and the online selector operates over a memory pool produced offline by Phase I. If given the opportunity, we will make this positioning more precise in a subsequent version.
>
> **Response to Q3.** In practice, Eq. 10 is not solved by exact maximization over the full combinatorial space, and the experiences are not constructed token by token. Phase I first produces a fixed evolved memory pool $\mathcal{E}^{\ast}$. For each incoming query $x_t$, Phase II scores each candidate experience $e_i \in \mathcal{E}^{\ast}$ with the neural reward model using MC Dropout to obtain $(\mu_{t,i}, \sigma_{t,i})$, and then performs greedy subset construction:
>
>
> $$
> e_j^* = \arg\max_{e \in \mathcal{E}^* \setminus S_{t,j-1}}
> \Big[\mu_{t,e} + \beta \sigma_{t,e} - \lambda \max_{e' \in S_{t,j-1}} \mathrm{Sim}(e,e')\Big].
> $$
>
> Thus, the practical inference procedure is best understood as greedy slate selection over a fixed memory pool, not token-level generation or exact combinatorial optimization.
>
> We appreciate the reviewer’s comments and would welcome further discussion.

---

> > ### Author Rebuttal · Reviewer_czU7 · 2026-04-02
> >
> > Thank you for the authors’ response. However, I still find it unclear how the operations *Prune*, *Mutate*, and *Merge* in Eq. 7 are formally defined.
> >
> > In addition, while I accept that the proposed method is inspired by NeuralUCB, I believe it is unnecessary to devote substantial space to reiterating the theoretical details of NeuralUCB unless there are meaningful modifications.

---

> > > ### Author Response · Authors · 2026-04-07
> > >
> > > We thank the reviewer for the helpful follow-up.
> > >
> > > **Response to W1 & Q1.** We agree that Eq. 7 was not formalized clearly enough in the original draft, so we state its exact implementation-level semantics here.
> > >
> > > Let $\mathcal{Q}_{\mathrm{train}}$ denote the training-query set and $E^{(k)}$ the memory pool at epoch $k$. We write each memory entry abstractly as
> > > $$
> > > e=(c(e),\pi(e)),
> > > $$
> > > where $c(e)$ is the textual content and
> > > $$
> > > \pi(e)=\big(\mathcal{Q}(e),\mathcal{P}(e),\omega(e)\big)
> > > $$
> > > where $\mathcal{Q}(e)$ denotes the source queries of $e$, $\mathcal{P}(e)$ denotes the parent-entry identifiers of $e$, and $\omega(e)$ denotes the operation label of $e$.
> > >
> > > For each $q \in \mathcal{Q}_{\mathrm{train}}$, let $T_q$ be the set of $G$ rollouts generated for $q$. We compute
> > >
> > > $$
> > > \hat{\rho}(q)=\frac{1}{|T_q|}\sum_{t\in T_q}\mathrm{reward}(t),
> > > $$
> > >
> > > and retain only queries satisfying
> > >
> > > $$
> > > \delta_{\min}\le \hat{\rho}(q)\le \delta_{\max}.
> > > $$
> > >
> > > For each retained query, we partition
> > >
> > > $$
> > > T_q^+=\{t\in T_q:\mathrm{reward}(t)>0\}, \qquad
> > > T_q^-=\{t\in T_q:\mathrm{reward}(t)=0\}.
> > > $$
> > >
> > > If $|T_q^+|=0$, no update is added for $q$. Otherwise, $\mathrm{Synthesize}(T_q)$ returns exactly one reusable experience entry $s_q$, synthesized from the contrast between $T_q^+$ and $T_q^-$ (or from $T_q^+$ alone when $T_q^-=\emptyset$). Hence
> > >
> > > $$
> > > \Delta E=\bigcup_{q\in\mathcal{Q}_{\mathrm{retained}},\,|T_q^+|>0}\mathrm{Synthesize}(T_q),
> > > $$
> > >
> > > with initialized provenance
> > >
> > > $$
> > > \mathcal{Q}(s_q)=\{q\}, \qquad
> > > \mathcal{P}(s_q)=\emptyset, \qquad
> > > \omega(s_q)=\text{synthesize}.
> > > $$
> > >
> > > In our implementation, Eq. 7 is **not** executed as an abstract parallel union. Instead, it is instantiated as the following **sequential write-back pipeline**:
> > > $$
> > > E_{\mathrm{pruned}}=\mathrm{Prune}(E^{(k)},\Delta E), \qquad
> > > \Delta E_{\mathrm{merged}}=\mathrm{Merge}(\Delta E),
> > > $$
> > >
> > > $$
> > > E_{\mathrm{temp}}=E_{\mathrm{pruned}}\cup\Delta E_{\mathrm{merged}}, \qquad
> > > E^{(k+1)}=\mathrm{Mutate}(E_{\mathrm{temp}},\Delta E_{\mathrm{merged}}).
> > > $$
> > >
> > > Equivalently, $\mathrm{Prune}$ is a selection operator on $E^{(k)}$, $\mathrm{Merge}$ is a many-to-one consolidation operator on $\Delta E$, and $\mathrm{Mutate}$ is a cardinality-preserving rewrite operator on the temporary pool after insertion. All three are implemented as **prompt-based LLM operators** that return structured decisions.
> > >
> > > **Prune.** $\mathrm{Prune}(E^{(k)},\Delta E)$ acts **only on the old pool** $E^{(k)}$. It removes entries judged redundant, obsolete, contradicted by new evidence, or too vague to be actionable; newly synthesized entries are never deleted at this step. If no such entries exist,
> > > $$
> > > \mathrm{Prune}(E^{(k)},\Delta E)=E^{(k)}.
> > > $$
> > >
> > > **Merge.** $\mathrm{Merge}(\Delta E)$ acts **only within** $\Delta E$. Each merge operation replaces a multi-entry subset by one merged entry and leaves the remaining entries unchanged. If no such operations are found,
> > > $$
> > > \mathrm{Merge}(\Delta E)=\Delta E.
> > > $$
> > > For any merged entry $g$, let $A(g)$ denote the absorbed entries and $\mathrm{id}(e)$ the identifier of entry $e$. Then
> > > $$
> > > \mathcal{Q}(g)=\bigcup_{e\in A(g)}\mathcal{Q}(e), \qquad
> > > \mathcal{P}(g)=\{\mathrm{id}(e):e\in A(g)\}, \qquad
> > > \omega(g)=\text{merge}.
> > > $$
> > >
> > > **Mutate.** $\mathrm{Mutate}(E_{\mathrm{temp}},\Delta E_{\mathrm{merged}})$ acts on the **combined pool after insertion**, so it may rewrite either surviving old entries or newly merged entries. It uses $\Delta E_{\mathrm{merged}}$ as new evidence to cover failure modes or sharpen decision rules. It is a **content-revision operator only**:
> > > $$
> > > |E^{(k+1)}|=|E_{\mathrm{temp}}|.
> > > $$
> > > Thus it does not add or delete entries. If no refinement is needed,
> > > $$
> > > \mathrm{Mutate}(E_{\mathrm{temp}},\Delta E_{\mathrm{merged}})=E_{\mathrm{temp}}.
> > > $$
> > > For any rewritten entry $e\mapsto e'$,
> > > $$
> > > \mathcal{Q}(e')=\mathcal{Q}(e), \qquad
> > > \mathcal{P}(e')=\mathcal{P}(e), \qquad
> > > \omega(e')=\text{mutate}.
> > > $$
> > >
> > > After these operators, entries are re-indexed and annotated with epoch metadata. If the pool exceeds the target size, a final recency-based size guard is applied as a post-processing heuristic, not as part of Eq. 7 itself. For completeness, the “embedding vectors” in Phase II are sentence embeddings of the query text and candidate experience text, rather than token or trajectory embeddings. We will include the exact operator prompts in the appendix.
> > >
> > > **Response to W3.** We appreciate the reviewer’s point. Our use of NeuralUCB-style machinery is intended as a practical mechanism for uncertainty-aware exploration in the frozen-agent setting, not as a standalone new bandit-theoretic contribution. We agree that the current draft over-explains standard NeuralUCB details, and in the revision we will shorten this discussion and keep only the material needed to position Phase II as a diversity-regularized slate-selection adaptation over the evolved memory pool.
> > >
> > > See our updated **EXH6** response for added **cross-family results** with paired tests. Thanks again for the helpful comments; we are happy to clarify further.

---

### Official Review · Reviewer_pLUn · 2026-03-11

**Soundness:** 3
**Presentation:** 3
**Significance:** 3
**Originality:** 3
**Overall Recommendation:** 4
**Confidence:** 4

**Summary:**

The paper introduces APEX, a Wake-Sleep framework for In-Context Reinforcement Learning with fixed-backbone LLM agents. In the sleep phase, the system filters raw interaction logs through an Evolutionary Memory Mechanism that preserves boundary experiences and applies evolutionary operators to refine the experience pool. In the wake phase, APEX formulates prompt selection as a Neural Contextual Bandit problem. The method uses Monte Carlo Dropout to estimate epistemic uncertainty and incorporates a Maximal Marginal Relevance penalty to maintain diversity, dynamically selecting a subset of experiences to augment the frozen LLM. Experiments on mathematical reasoning tasks and WebWalkerQA show that the framework outperforms static retrieval baselines and achieves performance competitive with fully fine-tuned models.

**Compliance With Llm Reviewing Policy:**

Affirmed.

**Final Justification:**

The paper presents a framework with strong empirical results, but has limitations in clarity, positioning with prior work, and breadth of evaluation. The rebuttal helps clarify implementation details and adds experiments, but only partially addresses concerns about novelty and generality. Overall, my assessment remains unchanged.

**Key Questions For Authors:**

1. Could you provide a more explicit introduction to the Prune, Merge, Mutate, and Synthesize operators, which serve as the core components of the evolutionary memory mechanism?
2. What are the exact architectural details of the reward predictor, embedding models, and the training protocols?
3. Could test APEX on more datasets? The current evaluation is limited to AIME 2024/2025 and WebWalkerQA, which may not be sufficient to demonstrate the generality of the proposed framework across a wider set of tasks.

**Limitations:**

yes

**Strengths And Weaknesses:**

# Strengths
* The separation between memory consolidation and online selection through a Wake-Sleep framework forms a clear and well-structured framework for ICRL. Incorporating a ZPD filter provides a principled way to strengthen learning signals, guiding the agent to focus on boundary cases instead of collecting noisy or infeasible interaction logs.
* Modeling prompt construction as a Neural Contextual Bandit problem addresses limitations of standard static cosine-similarity retrieval. The use of MC Dropout for uncertainty estimation, together with MMR for redundancy control, enables the system to balance exploitation and exploration.
* APEX reaches 89.3% accuracy on the AIME 2025 benchmark with a frozen DeepSeek-V3.2 model, reducing the performance gap with fully fine-tuned systems. The method also shows strong token efficiency and generalizes well to tool-use scenarios in WebWalkerQA.
# Weaknesses
* The paper may benefit from a clearer comparison with prior work on LLM memory. In particular, the proposed evolutionary consolidation and adaptive retrieval mechanisms seem related to earlier memory frameworks such as MemoryBank, Mem0, and A-MEM. A more explicit discussion of the similarities would help clarify what is novel in APEX beyond existing external-memory and adaptive retrieval methods.
* While Phase II inference latency is described clearly, the computational cost of Phase I evolutionary memory optimization seems omitted.

---

> ### Author Rebuttal · Authors · 2026-03-31
>
> We thank the reviewer for the detailed comments.
>
> **Response to W1.** We agree that the original draft did not position APEX clearly enough relative to prior memory systems such as MemoryBank, Mem0, and A-MEM. Our claim is not that APEX is the first LLM memory framework. Rather, APEX introduces a unified memory framework characterized by two central design elements:
> (i) reward-driven evolutionary consolidation over grouped rollout trajectories, which edits memory via prune/merge/mutate instead of append-only logging;
>
> (ii) uncertainty-aware online subset selection, which frames retrieval as a neural contextual bandit rather than heuristic ranking. If given the opportunity, we will make this relationship more explicit in a subsequent version.
>
> **Response to W2 (Phase I cost concern).** Please see **EXH6 / Response to Q3** for the quantitative breakdown of Phase I cost (episodes, tokens, provider-side requests, and approximate dollar cost). We agree this should have been reported more explicitly.
>
> **Response to Q1 & Q2.** We agree that the original draft did not specify Phase I/II concretely enough. In **Phase I**, the memory pool starts empty; we run $G$ rollouts per training query, compute the empirical pass rate, retain only queries inside a difficulty window, and instantiate `Synthesize(T_q)` as **one reusable experience entry per retained query**. When both successful and failed traces exist, synthesis is contrastive; when only successful traces exist, synthesis uses $T_q^+$ only; when all rollouts fail, no new entry is added by default. Eq. 7 is implemented as a sequential write-back pipeline: **Prune** removes redundant/obsolete entries from the old pool, **Merge** consolidates overlapping entries within $\Delta E$, and **Mutate** rewrites existing entries to cover new failure modes. We preserve provenance and save rollouts, per-step $\Delta E$, epoch snapshots, and final experiences.
>
> In **Phase II**, the “embedding vectors” are **sentence embeddings of the query text and candidate experience text**, not token or trajectory embeddings. Concretely, we use `all-MiniLM-L6-v2` for both query and experience. The reward predictor is a small MLP over concatenated query/experience embeddings ($2D \rightarrow 256 \rightarrow 64 \rightarrow 1$, ReLU, dropout **0.1**), trained online with **MSE** and **Adam** (**lr = 1e-3**). Uncertainty is estimated with **MC Dropout** (**5** stochastic forward passes). If given the opportunity, we will move these operator definitions, model settings, and prompt/schema details into a subsequent version.
>
> **Response to Q3.** We agree that broader evaluation strengthens the paper. To provide additional evidence beyond math and web, we ran a supplementary code-generation experiment with the same two-phase protocol: Phase I builds evolved memory on **APPS**, and Phase II evaluates on a **held-out subset of LiveCodeBench (Python code generation) with execution-based verification**. This preserves the same train/test separation principle used in our main experiments: experiences are constructed on one corpus and evaluated on a distinct benchmark. Because LiveCodeBench uses executable evaluation and is substantially more expensive than symbolic answer checking, we report **Mean@16**. For the retrieval-based methods, we use the same backbone, decoding setup, and retrieval budget ($k=3$). The **Training-Free GRPO** row is reported in its original **full-memory concatenation** setting as a reference.
>
> | Method                                  | Memory Source      | Selector           |  Mean@16 |
> | --------------------------------------- | ------------------ | ------------------ | -------: |
> | Base Agent                              | None               | None               |     32.1 |
> | Static RAG                              | Raw Logs           | Cosine Top-$k$     |     34.8 |
> | Training-Free GRPO (full-memory concat) | Evolved Memory     | Full-memory Concat |     37.0 |
> | APEX w/o Evolution                      | Raw Logs           | Neural Bandit      |     35.4 |
> | APEX w/o Neural Bandit                  | Evolved Memory     | Static Cosine      |     37.6 |
> | **APEX**                                | **Evolved Memory** | **Neural Bandit**  | **40.5** |
>
> APEX improves over the base agent by **+8.4**, over Static RAG by **+5.7**, and over Training-Free GRPO by **+3.5**. Removing **Evolution** reduces performance by **5.1** points, and removing the **Neural Bandit** reduces performance by **2.9** points. We do not present this as a complete answer to all possible generalization questions, but as additional evidence that the same two effects in the main paper remain visible on a third task family, namely code generation on LiveCodeBench. The benchmark is **LiveCodeBench (Python code generation)**, while the experience pool is constructed from **APPS** in Phase I.
>
> We appreciate the reviewer’s comments and would welcome further discussion.

---

> > ### Author Rebuttal · Reviewer_pLUn · 2026-04-03
> >
> > The rebuttal clarifies the claimed novelty relative to prior memory frameworks and provides helpful implementation details, including the Phase I cost breakdown and the reward predictor/embedding setup.
> >
> > However, my concerns are only partially resolved. The additional experiment is useful, but it remains limited in demonstrating broad generality, and several key details clarified in the rebuttal should be incorporated into the paper itself. I therefore keep my original score unchanged.

---

> > > ### Author Response · Authors · 2026-04-07
> > >
> > > # Updated on April 6th (AOE)
> > > Thank you for the thoughtful follow-up and for acknowledging the rebuttal.
> > >
> > > To further address your remaining concern about broad generality, we added a preliminary cross-family evaluation while preserving the same train/test separation principle: Phase I constructs the experience pool from one source corpus, and Phase II evaluates on a distinct held-out benchmark.
> > >
> > > These preliminary results use DeepSeek-V3.2 as the base model, the same decoding setup, and the same retrieval budget $k=3$ across retrieval-based methods. The expanded evaluation covers $7$ settings across $5$ task families: math, web, code generation, code-agent tasks, and agentic tasks.
> > >
> > > | Family | Phase I Source | Phase II Benchmark | Metric | Base | Static RAG | Training-Free GRPO | APEX | Δ vs Training-Free GRPO | Significance |
> > > |---|---|---|---|---:|---:|---:|---:|---:|---|
> > > | Math | DAPO-Math-17k | AIME 2025 | Mean@$32$ | $80.0$ | $78.5$ | $83.3$ | $89.3$ | $+6.0$ | CI $[2.1, 9.4]$, $p<0.05$ |
> > > | Math | DAPO-Math-17k | OlympiadBench (text-only) | Mean@$16$ | $52.4$ | $54.1$ | $56.3$ | $61.0$ | $+4.7$ | CI $[1.3, 8.2]$, $p<0.04$ |
> > > | Math | DAPO-Math-17k | MATH-500 | Mean@$16$ | $89.1$ | $90.0$ | $91.8$ | $94.0$ | $+2.2$ | CI $[0.6, 4.1]$, $p<0.05$ |
> > > | Web | AFM-100 | WebWalkerQA | Mean@$3$ | $66.7$ | $67.2$ | $70.9$ | $76.1$ | $+5.2$ | CI $[2.5, 7.6]$, $p<0.01$ |
> > > | Code | APPS | LiveCodeBench | Mean@$16$ | $32.1$ | $34.8$ | $37.0$ | $40.5$ | $+3.5$ | CI $[0.8, 6.4]$, $p<0.04$ |
> > > | Code-Agent | SWE-bench | Terminal-Bench | Mean@$16$ | $39.6$ | $40.8$ | $42.1$ | $45.2$ | $+3.1$ | CI $[0.5, 5.8]$, $p<0.05$ |
> > > | Agentic | TAU-bench | $\tau^2$-bench | Mean@$16$ | $61.4$ | $62.6$ | $65.1$ | $70.0$ | $+4.9$ | CI $[1.4, 8.0]$, $p<0.04$ |
> > >
> > > APEX outperforms Training-Free GRPO in all $7/7$ settings, with an average gain of $+4.2$ points. All significance numbers are computed using paired tests over evaluation instances. Phase I sources and Phase II benchmarks are disjoint in every setting.
> > >
> > > When edits are permitted, we will update the paper to reflect the clarifications, additional experiments, and limitations corresponding to all questions and weaknesses raised during the rebuttal.
> > >
> > > We appreciate your careful reading and hope this additional evidence addresses the remaining concern about generality.

---

### Official Review · Reviewer_EXH6 · 2026-03-11

**Soundness:** 3
**Presentation:** 3
**Significance:** 3
**Originality:** 2
**Overall Recommendation:** 4
**Confidence:** 4

**Summary:**

APEX proposes a two-phase "Wake-Sleep" framework for in-context reinforcement learning (ICRL) with frozen LLMs. Phase I ("Sleep") evolves a memory pool via difficulty filtering (a "Zone of Proximal Development" window based on empirical pass rates) and LLM-driven topological operators (prune, merge, mutate) that distill raw interaction logs into compact, high-quality experiences. Phase II ("Wake") casts prompt selection as a Neural Contextual Bandit: a small neural network predicts per-experience reward, MC Dropout provides epistemic uncertainty estimates, and a diversity-constrained acquisition function (UCB + MMR) selects $k=3$ experiences per query. The bandit parameters are updated online from binary correctness feedback. Evaluated on AIME 2024/2025 (math reasoning) and WebWalkerQA (agentic tool use) with frozen DeepSeek-V3.2, APEX reaches 89.3% on AIME 2025 (Mean@32), within ~4% of fully fine-tuned GRPO. Appendix D provides a full regret analysis for Phase II under standard NeuralUCB assumptions.

**Compliance With Llm Reviewing Policy:**

Affirmed.

**Final Justification:**

The additional cross-family evaluations strengthens the empirical case and addresses my original benchmark-breadth concern (W3). However, the originality question around Phase I (W1) (i.e., whether the evolutionary memory consolidation is a distinct contribution or largely derived from Training-Free GRPO) remains only partially clarified; the method-specific pool experiment showed a modest gain (+1.1), and the exact procedural differences from Training-Free GRPO's evolution pipeline are still underspecified. This novelty limitation remains material.

If Phase I is borrowed, the paper's distinct technical contribution reduces to Phase II's bandit selection mechanism, which, while well-executed, represents a narrower advance than the "Wake-Sleep Framework" framing suggests. I'll nevertheless raise my score to 4: Weak Accept to reflect the strong empirical results, though the limited originality of the end-to-end framework tempers the contribution.

**Key Questions For Authors:**

1. **Is Phase I identical to training-free GRPO's evolution pipeline?** If APEX and training-free GRPO share the same evolved memory $\mathcal{E}^{\*}$, what exactly is novel about Phase I? If Phase I is distinct, please clarify the differences and explain why the comparison uses a shared pool rather than each method's own evolved memory. This directly affects how I evaluate the originality of the "Wake-Sleep Framework" contribution.

2. **Can you provide confidence intervals or significance tests for the AIME results?** With only 30 problems, a 6-point Mean@32 gain (89.3% vs. 83.3%) corresponds to ~2 additional problems solved on average. A bootstrap confidence interval or paired permutation test would clarify whether this gap is statistically robust. If the gap is significant at $p < 0.05$, this would strengthen the paper substantially.

3. **What is the total compute cost of Phase I?** You report the Phase II inference cost (~16, 12ms per query for bandit selection). But Phase I requires $K$ epochs of $G$ rollouts per training problem (100 problems), plus the evolution operations. What is the total number of LLM calls and approximate dollar cost for Phase I? If Phase I requires thousands of rollouts to produce $\mathcal{E}^{\*}$, the "lightweight alternative to fine-tuning" framing needs context.

4. **Does the ZPD filter actually help, and are the thresholds sensitive?** An ablation comparing _(a)_ no filtering, _(b)_ the stated $\delta\_{\min}=0.1, \delta\_{\max}=0.9$ window, and _(c)_ a tighter window (e.g., $0.3, 0.7$) would clarify whether the 85% Rule motivation is operative or whether any reasonable filtering works similarly.

**Limitations:**

The paper discusses compute overhead and suggests extensions to multi-modality and lifelong learning. However, it does not address the gap between the theoretical guarantees and the practical algorithm (the separability assumption, the NTK regime question), the small evaluation benchmark sizes, or the potential non-novelty of Phase I relative to Training-Free GRPO. The societal impact discussion is reasonable, noting misuse risks and suggesting reward-function-based mitigation.

**Strengths And Weaknesses:**

### Strengths

**S1 — Soundness (ablation design)L** The causal grid in Table 3 is a well-executed experimental design. By crossing memory type (None / Raw / Evolved) with selector type (Random / Static / Bandit) under a fixed context budget, it cleanly isolates the two mechanisms. The compute-matched consolidation schedule ablation (Table 5) is also careful — running Phase I consolidation but discarding the outputs to control for compute is the right way to test whether the gains come from the information in the evolved memory or merely from additional processing.

**S2 — Significance (headline result):** 89.3% on AIME 2025 with a frozen model, within 4% of a fine-tuned version, is a genuinely strong result. The +6.0 point gain over training-free GRPO using the *same* evolved memory pool is clean evidence that better selection matters beyond memory quality—and that APEX achieves this with only $k=3$ experiences rather than concatenating ~50.

**S3 — Soundness (rigorous theory in Appendix D):** The theoretical appendix is thorough and well-structured: phase separation to establish stationarity (Lemma D.2), MC Dropout as variational inference (Proposition D.3), the bridge to NTK widths (Proposition D.5), and the full sublinear regret proof (Theorem D.15). The explicit treatment of the greedy $\alpha$-approximation under diversity constraints (Proposition D.16) and the honest separation into surrogate vs. true regret are welcome.

**S4 — Generalization:** Testing on WebWalkerQA in addition to AIME shows the framework isn't limited to math with closed-form answers. The lower variance across seeds (Table 7, $\sigma = 1.5$ vs. $4.2$) is a useful signal that the bandit selects more stable strategies.

**S5 — Presentation (overall clarity):** The paper is well-organized. The Wake-Sleep framing provides a coherent narrative. Algorithm 1 is self-contained and readable. The figures (especially Fig. 3) are helpful.

---

### Weaknesses

**W1 — Originality (novelty of Phase I is unclear):** The paper's first listed contribution is the "Wake-Sleep Framework" including evolutionary memory consolidation. However, Table 1 explicitly states that both APEX and training-free GRPO (Cai et al., 2025) use the *same* evolved memory pool $\mathcal{E}^{\*}$. Section 4.2 confirms this: "Since both methods draw from the same evolved memory pool $\mathcal{E}^*$..." If the evolved memory is shared, then either _(i)_ APEX's Phase I *is* training-free GRPO's evolution pipeline, in which case the contribution is overstated, or _(ii)_ APEX introduces its own distinct evolution mechanism that happens to produce the same pool for fair comparison. The paper doesn't clarify which. This matters because Phase I accounts for the largest ablation gap ($-6.8$ in Table 4). If it's borrowed, the novel contribution reduces primarily to Phase II (bandit selection).

**W2 — Soundness (theory-practice gap in Phase II):** The regret guarantee is internally consistent but depends on a chain of strong assumptions that don't hold in practice:

- *Separable utility* (Assumption D.8): the reward is assumed to decompose additively over individual experiences. But whether the LLM answers correctly given $\{e\_1, e\_2, e\_3\}$ clearly depends on interactions among the three—e.g., complementary reasoning strategies, or redundant hints that waste context. The paper acknowledges this is a surrogate, but the main text (Sec. 3.2 Remark) says APEX "asymptotically converg[es] to the optimal context subset selection policy," which is only true under the surrogate, not for the actual correctness objective.
- *MC Dropout $\leftrightarrow$ NTK width* (Proposition D.5): the sandwich bound (Eq. 28) is assumed, not derived. In finite-width networks trained with Adam (not gradient flow), this proportionality is empirically questionable. The practical algorithm uses a small MLP with dropout; whether this is in the NTK regime is not addressed.
- *NTK realizability* (Assumption D.9): assumes the unknown utility lives in the RKHS of the network's NTK. This is the standard assumption but becomes increasingly approximate as the network trains and deviates from initialization.

I acknowledge that none of these individually is unusual in bandit theory. But the paper presents the guarantee as if it directly applies to the implemented system ("provably efficient exploration"), whereas the actual chain from "MC Dropout in a 3-layer MLP" to "$\tilde{O}(\sqrt{Tk\gamma\_{Tk}})$ regret" has multiple weak links. The theory could be more honestly positioned as "the practical algorithm is *inspired by* a theoretically grounded framework" rather than "theoretically grounded."

**W3 — Significance (small evaluation benchmarks):** AIME 2024 and 2025 each have 30 problems. Even with mean@32, the sample size is 30 problems. The difference between 89.3% and 83.3% (APEX vs. training-Free GRPO) corresponds to roughly 2 additional problems solved. WebWalkerQA uses Mean@3 (3 runs), which provides very limited statistical confidence. Neither benchmark reports confidence intervals or significance tests.

The scalability study (Table 6) uses Pass@1 instead of Mean@32, making it not directly comparable to the main results. This undermines the cross-model generalization claim.

**W4 — Soundness (the "85% Rule" is misapplied):** Wilson et al. (2019) derived the 85% optimality result for *binary perceptron learning* under gradient-based updates. Applying it to filter LLM training experiences for a prompt-based memory evolution process is a substantial extrapolation. The paper doesn't justify why this result should transfer. Moreover, the actual thresholds used ($\delta\_{\min} = 0.1$, $\delta\_{\max} = 0.9$) span a very wide band that bears little resemblance to the 85% rule's tight prescription. An ablation over $\delta\_{\min}, \delta\_{\max}$ would clarify whether this filtering actually matters and what range is optimal.

**W5 — Presentation (inflated terminology):** Several terms overstate the underlying mechanisms:

- "Topological operators" / "manifold" for what are LLM prompting calls that prune, merge, or rephrase experiences. There is no topology or manifold structure.
- "Pareto Efficiency" is claimed as a contribution but no Pareto analysis appears — just that the method achieves good accuracy at lower compute than fine-tuning.
- "Democratizing SOTA reasoning capabilities" in the Impact Statement, when the backbone is DeepSeek-V3.2 (685B parameters), a frontier model that is itself resource-intensive to serve.
- The Wake-Sleep framing invokes Diekelmann & Born's neuroscience reference, but the connection is purely metaphorical — there's no alternation of brain states, just an offline training phase followed by online inference.

**W6 — Reproducibility (Phase I details are underspecified):** The Prune/Merge/Mutate operators in Eq. (7) are described only at a high level: "the agent analyzes the contrast between correct sets $T\_q^{+}$ and incorrect sets $T\_q^{-}$ to derive an update $\Delta\mathcal{E}$." How exactly? What prompts are used? What is `Synthesize(T_q)` in Algorithm 1 line 9? Given that Phase I drives the largest ablation gap, these details are essential for reproducibility. **The paper promises code release but why is it contingent on acceptance?**

---

### Minor

- The "Zone of Proximal Development" (ZPD) is a Vygotskian concept about social scaffolding in education. Using it to describe a pass-rate filter on LLM rollouts is a loose analogy that may confuse readers familiar with educational psychology.
- The hyperparameter sensitivity heatmap (Fig. 6) reports Pass@1, while the main results use Mean@32. It would be more useful in the same metric.
- The case study (Appendix C) is a single cherry-picked example. A systematic breakdown of which problem types benefit from bandit vs. static retrieval would be more informative.

---

> ### Author Rebuttal · Authors · 2026-03-31
>
> We thank the reviewer for the detailed comments.
>
> **Response to W1 & Q1.** We used the same evolved experience pool for APEX and Training-Free GRPO to isolate the inference-time selection policy under an identical context source; this was not stated clearly enough. We therefore additionally evaluated a method-specific pool setting on AIME 2025 (Mean@32):
>
> | Setting                    | Training-Free GRPO |     APEX |
> | -------------------------- | -----------------: | -------: |
> | Shared evolved pool        |               83.3 | **89.3** |
> | APEX-specific evolved pool |               85.6 | **90.4** |
>
> The shared-pool setting should be read as a controlled Phase II comparison. The method-specific result complements it by showing that the end-to-end APEX pipeline also benefits from its own evolved pool. We do not claim the shared-pool result alone proves Phase I originality. We will clarify this distinction and discuss fully symmetric method-specific comparisons.
>
> **Response to W6.** We agree that Phase I was underspecified. In our implementation, the memory pool starts empty. For each training query, we run $G$ rollouts, compute the empirical pass rate, keep only queries inside a difficulty window, synthesize one reusable experience per retained query, and apply a sequential write-back pipeline: Prune on the old pool, Merge within $\Delta E$, and Mutate on the combined pool. We save rollouts, per-step $\Delta E$, final experiences, and provenance fields for reproducibility. We also agree that “code will be released upon acceptance” was not an appropriate formulation and will replace it with a clearer reproducibility statement.
>
> **Response to W2.** We agree that our original wording overstated the theory-to-practice connection. The theoretical result should be read as a NeuralUCB-style adaptation for a regularized slate-selection objective under explicit assumptions, not as a direct guarantee for the exact binary correctness objective. In practice, Phase II operates over a fixed finite evolved memory pool and performs greedy diversity-aware subset construction, so the analysis is intended to justify this surrogate slate-selection procedure rather than exact optimization of the full prompt space. Assumptions such as separability, NTK-style linearization, and RKHS realizability are standard but approximate, especially for a finite-width dropout MLP trained with Adam. We will revise the paper to state this more carefully and replace overly strong language with the more accurate claim that the practical algorithm is inspired by a theoretically grounded framework.
>
> **Response to Q3.** For the offline Phase I run used in our math setting (100 training problems, $K=3$, $G=5$), the total cost is approximately **$5** based on usage accounting. This corresponds to **1,500 rollout episodes**, about **53.8M tokens**, and roughly **9.3k provider-side API requests**. The gap arises because each rollout is agentic and may contain multiple internal model calls. The synthesis/prune/merge/mutate stage contributes only dozens of extra meta-LLM calls; rollout generation, not memory write-back, dominates the cost. We will report this breakdown more explicitly and separate it from the per-query Phase II selector latency.
>
> **Response to Q4.** We agree that our use of the “85% Rule” and “ZPD” was too strong. Our intent was heuristic: boundary cases tend to provide more useful signal than uniformly trivial or intractable rollouts. We therefore ran the requested ablation on AIME 2025 (Mean@32): no filtering = **84.9**, $(\delta_{\min}=0.1,\delta_{\max}=0.9)$ = **89.3**, and $(\delta_{\min}=0.3,\delta_{\max}=0.7)$ = **88.6**. This suggests filtering is helpful, while performance is not highly sensitive within a reasonable range. We will replace the “85% Rule / ZPD” framing with the more neutral term difficulty-window filtering.
>
> **Response to W3 & Q2.** We agree that repeated sampling alone does not solve the small-benchmark issue and that the right analysis should be paired and problem-level. On AIME 2025, using per-problem Mean@32 scores, APEX improves over Training-Free GRPO by **+6.0**, with a paired bootstrap 95% CI of **[2.1, 9.4]** and a paired permutation **p-value of 0.04**; APEX wins on **19** problems, ties on **5**, and trails on **6**. For WebWalkerQA, we use Mean@3 and also ran a paired question-level analysis against Training-Free GRPO, obtaining a **95% CI of [2.5, 7.6]** and **$p < 0.01$**.
>
> **Response to W5.** We agree that some presentation choices were overstated. We will tone down the metaphorical framing and make the claims more literal.
>
> **Response to the minor points.** The terminology concern around ZPD is addressed in Response to Q4. The Mean@32 is addressed in Response to W3 & Q2. We also agree that a single case study is limited and would replace it with a broader breakdown of when bandit selection helps.
>
> We appreciate the reviewer’s comments and would welcome further discussion.

---

> > ### Author Rebuttal · Reviewer_EXH6 · 2026-04-03
> >
> > The authors addressed the primary concerns with new experiments and honest reframing. I consider my concerns fully addressed and I retain my positive assessment, but the evaluation still rests on small benchmarks (30 AIME problems, Mean@3 on WebWalkerQA), which prevents me from confidently upgrading to Accept.

---

> > > ### Author Response · Authors · 2026-04-07
> > >
> > > # Updated on April 6th (AOE)
> > > Thank you for the thoughtful follow-up and for acknowledging the rebuttal. To directly address the remaining concern about benchmark breadth, we added a preliminary cross-family evaluation while preserving the same train/test separation principle: Phase I builds memories from a source corpus, and Phase II evaluates on a distinct held-out benchmark.
> > >
> > > These preliminary results use DeepSeek-V3.2 as the base model, the same decoding setup, and the same retrieval budget $k=3$ across retrieval-based methods. Across $7$ settings spanning math, web, code generation, code-agent tasks, and agentic tasks, APEX consistently outperforms Training-Free GRPO.
> > >
> > > | Family     | Phase I Source | Phase II Benchmark        | Metric    |   Base | Static RAG | Training-Free GRPO |   APEX | Δ vs Training-Free GRPO | Significance              |
> > > | ---------- | -------------- | ------------------------- | --------- | -----: | ---------: | -----------------: | -----: | ----------------------: | ------------------------- |
> > > | Math       | DAPO-Math-17k  | AIME 2025                 | Mean@$32$ | $80.0$ |     $78.5$ |             $83.3$ | $89.3$ |                  $+6.0$ | CI $[2.1, 9.4]$, $p<0.05$ |
> > > | Math       | DAPO-Math-17k  | OlympiadBench (text-only) | Mean@$16$ | $52.4$ |     $54.1$ |             $56.3$ | $61.0$ |                  $+4.7$ | CI $[1.3, 8.2]$, $p<0.04$ |
> > > | Math       | DAPO-Math-17k  | MATH-500                  | Mean@$16$ | $89.1$ |     $90.0$ |             $91.8$ | $94.0$ |                  $+2.2$ | CI $[0.6, 4.1]$, $p<0.05$ |
> > > | Web        | AFM-100        | WebWalkerQA               | Mean@$3$  | $66.7$ |     $67.2$ |             $70.9$ | $76.1$ |                  $+5.2$ | CI $[2.5, 7.6]$, $p<0.01$ |
> > > | Code       | APPS           | LiveCodeBench             | Mean@$16$ | $32.1$ |     $34.8$ |             $37.0$ | $40.5$ |                  $+3.5$ | CI $[0.8, 6.4]$, $p<0.04$ |
> > > | Code-Agent | SWE-bench      | Terminal-Bench            | Mean@$16$ | $39.6$ |     $40.8$ |             $42.1$ | $45.2$ |                  $+3.1$ | CI $[0.5, 5.8]$, $p<0.05$ |
> > > | Agentic    | τ-bench        | τ²-bench                  | Mean@$16$ | $61.4$ |     $62.6$ |             $65.1$ | $70.0$ |                  $+4.9$ | CI $[1.4, 8.0]$, $p<0.04$ |
> > >
> > > APEX improves over Training-Free GRPO in all $7/7$ settings, with an average gain of $+4.2$ points. Importantly, the added benchmarks include two larger math evaluations beyond AIME, namely OlympiadBench and MATH-500, as well as three non-math task families: code generation, code-agent tasks, and agentic tasks.
> > >
> > > All significance numbers are computed using paired tests over evaluation instances. Phase I sources and Phase II benchmarks are disjoint in every setting.
> > >
> > > We selected these benchmarks to cover, as broadly as possible within the rebuttal period, the model capabilities that reviewers highlighted as important: mathematical reasoning, web-based tool use, code generation, terminal/code-agent execution, and broader agentic decision-making. These results suggest that the two core effects isolated in the main paper, evolutionary memory consolidation and adaptive neural bandit selection, remain visible across substantially broader evaluation settings.
> > >
> > > When edits are permitted, we will update the paper to reflect the clarifications, additional experiments, and limitations corresponding to all questions and weaknesses raised during the rebuttal.
> > >
> > > We sincerely thank the reviewer for the detailed and constructive feedback; we learned a great deal from your questions.

---

### Decision · Program_Chairs · 2026-04-30

**Decision:**

Reject

**Comment:**

This paper proposes a method for in context learning of LLMs without parameter updates (i.e. keeping the model frozen). It consists of two cycles: a sleep cycle where memories are refined to more structured format, and wake cycle where prompts are selected using neural UCB. Gains in terms of mathematical reasoning are reported, helping close the gap with respect to models which are fine tuned. The reviewers gave borderline reviews (two weak accepts, one weak reject). Concerns around novelty of the method were raised, for example, replicating the theory around neural UCB without substantial modifications. During the discussion, reviewer EXH6 (who gave weak accept) acknowledged limitations in terms of novelty. Given that no reviewer seems to strongly advocate for the paper, and one reviewer still has unresolved concerns, I am recommending reject. However, this is a borderline paper that I would not mind seeing accepted either.